# Historical and future anthropogenic warming effects on droughts, fires and fire emissions of $CO_2$ and $PM_{2.5}$ in equatorial Asia when 2015-like El Niño events occur

Hideo Shiogama[1, 2], Ryuichi Hirata[1], Tomoko Hasegawa[3], Shinichiro Fujimori[4], Noriko Ishizaki[1], Satoru Chatani[1], Masahiro Watanabe[2], Daniel Mitchell[5], Y. T. Eunice Lo[5]

[1]National Institute for Environmental Studies, 16-2 Onogawa, Tsukuba, Ibaraki 305-8506, Japan
[2]Atmosphere and Ocean Research Institute, University of Tokyo, 5-1-5 Kashiwanoha, Kashiwa, Chiba 277-8564, Japan
[3]College of Science and Engineering, Ritsumeikan University, 1-1-1 Noji-higashi, Kusatsu, Shiga 525-8577, Japan
[4]Department Environmental Engineering, Graduate School of Engineering, Kyoto University, Kyoto 615-8540 Japan
[5]School of Geographical Sciences, University of Bristol, University Road, Bristol BS8 1SS, United Kingdom

*Corresponding author*: Hideo Shiogama (shiogama.hideo@nies.go.jp)

**Abstract.** In 2015, El Niño contributed to severe droughts in equatorial Asia (EA). The severe droughts enhanced fire activities in the dry season (June-November), leading to massive fire emissions of $CO_2$ and aerosols. Based on large event attribution ensembles of the MIROC5 atmospheric global climate model, we suggest that historical anthropogenic warming increased the chances of meteorological droughts exceeding the 2015 observations in the EA area. We also investigate changes in drought in future climate simulations, in which prescribed sea surface temperature data have the same spatial patterns as the 2015 El
Niño with long-term warming trends. Large probability increases of stronger droughts than the 2015 event are projected when events like the 2015 El Niño occur in the 1.5°C and 2.0°C warmed climate ensembles according to the Paris Agreement goals. Further drying is projected in the 3.0°C ensemble according to the current mitigation policies of nations.

We use observation-based empirical functions to estimate burned area, fire $CO_2$ emissions and fine (<2.5 µm) particulate matter ($PM_{2.5}$) emissions from these simulations of precipitation. There are no significant increases in the chances of burned
areas and the $CO_2$ and $PM_{2.5}$ emissions exceeding the 2015 observations due to past anthropogenic climate change. In contrast, even if the 1.5°C and 2.0°C goals are achieved, there are significant increases in the burned area and $CO_2$ and $PM_{2.5}$ emissions. If global warming reaches 3.0°C, as is expected from the current mitigation policies of nations, the chances of burned area, $CO_2$ and $PM_{2.5}$ emissions exceeding the 2015 observed values become approximately 100%, at least in the single model ensembles.

We also compare changes in fire $CO_2$ emissions due to climate change and the land-use $CO_2$ emission scenarios of five shared socioeconomic pathways, where the effects of climate change on fire are not considered. There are two main implications. First, in a national policy context, future EA climate policy will need to consider these climate change effects regarding both mitigation and adaptation aspects. Second, the consideration of fire increases would change global $CO_2$ emissions and the mitigation strategy, which suggests that future climate change mitigation studies should consider these factors.

## 1 Introduction

El Niño events, often characterized through their positive sea surface temperature (SST) anomalies in the central and eastern tropical Pacific Ocean, accompany a weakening of the Walker circulation in the equatorial Pacific region. In the equatorial Asia region (EA, the area denoted in Fig. 1g), the weakening of the Walker circulation due to major El Niño events corresponds to downward motion anomalies and less convection (negative precipitation anomalies) (Santoso et al., 2017). The 2015/2016
major El Niño event (strongest since 1997/1998) induced negative precipitation anomalies and enhanced severe drought in the

EA region during the dry season (June-November) of 2015 (Field et al., 2016; Liu et al., 2017; Santoso et al., 2017). Parts of the EA region are tropical peatlands that contain tremendous amounts of soil organic carbon (Page et al., 2011) and huge biomass (Baccini et al., 2012, 2017; Saatchi et al., 2011). Coupled with anthropogenic land-use change (e.g., expansion of oil palm plantations on peatlands), severe drought increased fire activities in forests and peatlands, leading to large economic loss (at least 16.1 billion USD for Indonesia) and significant impacts on ecology and human health (Taufik et al., 2017; World Bank 2016, Hartmann et al., 2018). The fires enhanced the emissions of $CO_2$ and aerosols (Yin et al., 2016; Field et al., 2016; Koplitz et al., 2016; Stockwell et al., 2016; Liu et al., 2017). The fire carbon emissions of 2015 were the largest since the 1997 El Niño year (Yin et al., 2016). The estimated 2015 $CO_2$-equivalent biomass burning emissions for all Indonesia (1.5 billion metric tons $CO_2$) were between the 2013 annual fossil fuel $CO_2$ emissions of Japan and India (Field et al., 2016). The massive emissions of ozone precursors and aerosols, including fine (<2.5 micrometers) particulate matter ($PM_{2.5}$), caused severe haze across much of EA (Field et al., 2016), resulting in the excess deaths of approximately 100,300 people (Koplitz et al., 2016).

In a previous study (Lestari et al., 2014), we suggested that recent fire events in Sumatra were exacerbated by human-induced drying trends based on analyses of two sets of historical simulations of the MIROC5 atmospheric global climate model (AGCM) (Watanabe et al., 2010) with and without anthropogenic warming. Lestari et al. (2014) and Yin et al. (2016) projected future increases in the frequencies of droughts and fires based on analyses of the coupled atmosphere-ocean global climate model (AOGCM) ensembles of the Coupled Model Intercomparison Project Phase 5 (CMIP5) (Taylor et al., 2012).

Although Lestari et al. (2014) showed the anthropogenic effects on the *historical trends in droughts,* it is not clear how historical climate change affected the *particular drought event of 2015.* Because extreme events can occur by natural variability alone, it is difficult in principle to attribute a particular event to anthropogenic climate change. However, comparisons of observations and large ensemble simulations can help us evaluate the degree to which human influence has affected the probability of a particular event (Allen 2003). Such an approach is called probabilistic event attribution (PEA) (Pall et al. 2011, Shiogama et al. 2013). In the PEA approach, two sets of large ensemble (e.g., 100 members) are generally performed. The first are historical simulations of an AGCM driven by the historical values of anthropogenic (e.g., greenhouse gases) and natural forcing (solar and volcanic activities) agents and by the observed SST and sea ice concentration (SIC). The second is counterfactual natural runs driven by preindustrial anthropogenic and historical natural forcing agents and by the observed values of SST and SIC cooled according to estimates of anthropogenic warming (Stone et al. 2019) (see section 3 for more details). Note that the components of interannual variations in the SST data are not modified in the natural forcing ensemble. Therefore, for example, we can assess how anthropogenic warming affected the probabilities of drought events exceeding the observed value in the 2015 major El Niño year by comparing the distributions of members in historical and natural forcing ensembles. In this study, based on the PEA approach, we examine whether historical climate change increased not only the probabilities of drought but also those of fire and fire emissions of $CO_2$ and $PM_{2.5}$ during the June-November dry season of 2015. The lower computing costs of AGCM than AOGCM enable us to perform large ensembles, which are necessary for PEA. We use the 100-member PEA ensembles of MIROC5 (Shiogama et al. 2014) that have been used for many attribution studies of single extreme events (e.g., Shiogama et al., 2014, Kim et al., 2018, Hirota et al., 2018).

Although Lestari et al. (2014) and Yin et al. (2016) showed increases in droughts and fires in the future *transient* projection ensembles of AOGCMs, it is not clear how future anthropogenic warming affects droughts and fire when events like the 2015 El Niño occur in a future warmer climate. It is also important to investigate changes in extreme events at 1.5°C and 2.0°C warming levels to inform stakeholders after that the Paris Agreement set the 2°C long-term climate stabilization goal and moreover state pursuing 1.5 °C for stabilization (United Nations Framework Convention on Climate Change 2015). In this study, we examine how the probabilities of drought, fire and fire emissions of $CO_2$ and $PM_{2.5}$ would change when major El Niño events like 2015 occur under 1.5°C and 2.0°C warmed climates. We analyse large (100-member) ensembles of the MIROC5 AGCM under the Half a degree Additional warming, Prognosis and Projected Impacts (HAPPI) project, which was initiated in response to the Paris agreement (Mitchell et al., 2016, 2017, 2018; Shiogama et al., 2019). These MIROC5 HAPPI

ensembles have been used, for example, to study the changes in extreme hot days (Wehner et al., 2018), extreme heat-related mortality (Mitchell et al., 2018), tropical rainy season length (Saeed et al., 2018) and global drought (Liu et al., 2018) at 1.5℃ and 2.0℃ global warming. There is a significant "emissions gap", which is the gap between where we are likely to be and where we need to be (United Nations Environment Programme 2018). The current mitigation policies of nations would lead to global warming of approximately 3.2°C (with a range of 2.9-3.4°C) by 2100 (United Nations Environment Programme 2018). Therefore, it is worthwhile to compare changes in extreme events and impacts in cases where the 1.5℃ and 2.0℃ goals are achieved or not. Therefore, we perform and analyse a large ensemble of a 3.0℃ warmed climate.

By using the above ensembles, we answer the following questions:

(a) Has historical climate change significantly affected the probabilities of drought, fire and fire emissions of $CO_2$ and $PM_{2.5}$?

(b) How do the probabilities of drought, fire and fire emissions in 2015-like major El Niño years change if we can limit global warming to 1.5℃ and 2.0℃? Adaptation investments are necessary to reduce the associated impacts.

(c) If we overshoot the 1.5℃ and 2.0℃ goals to the current trajectory of 3.0℃, how will drought, fire and fire emissions be altered? Comparisons of the results of 3.0℃ and 2.0℃/1.5℃ indicate the potential benefits of mitigation efforts to achieve the goals of the Paris Agreement.

Although conversions of forest and peatlands to agriculture and plantations of oil palm are also important factors for fire activities (Marlier et al., 2013, 2015; Kim et al., 2015), we do not examine the effects of land use change in this study. In sections 2 and 3, we describe the empirical functions and model simulations used in this study, respectively. In section 4, we examine changes in precipitation, fire and fire emissions. Finally, section 5 contains the conclusions.

## 2 Empirical functions

Figures 1a-c indicate the observed June-November 2015 mean anomalies in surface air temperature ($\triangle T$), vertical pressure velocity at the 500-hPa level ($\triangle\omega_{500}$) and precipitation ($\triangle P$) relative to the 1979-2016 averages. ERA Interim reanalysis (ERA-I) data (Dee et al., 2011) are used for $\triangle T$ and $\triangle\omega_{500}$. Global Precipitation Climatology Project (GPCP) data (Adler et al., 2003) are analysed for $\triangle P$. The largely positive $\triangle T$ over the eastern tropical Pacific Ocean (i.e., El Niño) is related to substantial downward motion anomalies (weakening of Walker circulation) and negative precipitation anomalies over the EA region (the area shown in Fig. 1g). The negative precipitation anomalies in June-November 2015 were the third largest since 1979 (the first and second largest anomalies are the 1997 and 1982 El Niño year).

In the EA region, the negative precipitation anomalies are associated with the enhanced fire fraction, fire $CO_2$ emissions and fire $PM_{2.5}$ emissions estimated from the Global Fire Emissions Database (GFED4s) (van der Werf et al., 2017) (Figs. 1d-f). By combining satellite information on fire activity and vegetation productivity, GFED4s provide monthly burned area, fire carbon and dry matter (DM) emissions. We can also compute aerosol emissions by multiplying DM by the provided factors. The $CO_2$ and $PM_{2.5}$ emissions increase linearly as the burned areas expand (Supplementary Fig. 1). Previous studies found that fire activities and related emissions have non-linear relationships with precipitation anomalies and accumulated water deficits (Lestari et al., 2014; Spessa, et al. 2015; Yin et al., 2016; Field et al., 2016). Figure 2 shows the empirical relationships between the EA averaged precipitation anomalies (GPCP) and the EA cumulative burned area and fire $CO_2$ and $PM_{2.5}$ emissions (GFED4s) during 1997-2016. Here, we remove the 1979-2016 average from precipitation and divide the anomalies by their standard deviation value. As precipitation decreases, the burned area, fire $CO_2$ and $PM_{2.5}$ emissions increase exponentially. We estimate the fitting curves (solid curves in Fig. 2) by using the following equation:

$$\ln(y) = a + b\Delta P, \qquad\qquad (Eq. 1)$$

where $y$ is the burned area, $CO_2$ emissions or $PM_{2.5}$ emissions, and $a$ and $b$ are the intercept and regression coefficient, respectively. The coefficients of determination ($R^2$) are higher than 0.7. We also estimate the 10%-90% confidence intervals of the fitting curves by applying a 1000-time random sampling of the observed data: we randomly resample 20-year samples from the original 20-year (1997-2016) data and compute $a$ and $b$; we repeat the random resampling process 1000-times; we consider that the 10%-tile and 90%-tile values of the 1000 regression lines indicate the 10%-90% confidence intervals. These non-linear relationships are consistent with previous studies (Lestari et al., 2014; Spessa, et al. 2015; Yin et al., 2016; Field et al., 2016). We use the relationships in Figs. 2a-c as empirical functions to estimate burned area and fire emissions from the AGCM simulations of precipitation in section 4.

## 3 Model simulations

The MIROC5 AGCM (Watanabe et al. 2010) has a 160 km horizontal resolution. We perform 10-member long-term (1979-2016) historical simulations (Hist-long) of the MIROC5 AGCM forced by the observed sea surface temperature (SST) (HadISST, Rayner et al., 2003) and anthropogenic and natural external forcing factors (Shiogama et al., 2013; 2014). Here, the observed $\triangle P$ and $\triangle \omega_{500}$ are divided by their standard deviation values. The $\triangle P$ and $\triangle \omega_{500}$ of each ensemble member are also divided by their own standard deviation values. The correlations of the 1979-2016 time series of $\triangle P$ and $\triangle \omega_{500}$ between the observations and the ensemble averages of the MIROC5 simulations are 0.90 and 0.87, respectively (Figs. 3a-b). When we apply the above normalization process as a simple bias correction technique, it is found that the MIROC5 model has good hindcast skill regarding interannual variability in the EA-averaged $\triangle P$ and $\triangle \omega_{500}$. The precipitation and vertical motion anomalies are closely related to the Nino 3.4 SST (an index of El Niño Southern Oscillation) in the observations (correlations are -0.89 and 0.76, respectively) (Figs. 3c-d). There is also a high correlation value between $\triangle P$ and $\triangle \omega_{500}$ (-0.87) (Fig. 3e). We show that El Niño (La Niña) accompanies descending wind (ascending wind) in the EA area (Fig. 3d), leading to negative (positive) $\triangle P$ (Figs. 3e and 3c). The MIIROC5 model well represents these relationships between Niño 3.4, $\triangle P$ and $\triangle \omega_{500}$ in the observations (Figs. 3c-e), i.e., the regression lines of MIROC5 in Figs. 3c-e are close to those in the observations.

To investigate whether historical anthropogenic climate change affected the precipitation anomalies during the 2015 El Niño event, we analyse the outputs of two large ensembles, one with factual historical forcing (Hist) and one with counterfactual natural forcing (Nat) of MIROC5 for June-November 2015 (Shiogama et al. 2013, 2014). These simulations are called "probabilistic event attribution" experiments, which contribute to "the International Climate and Ocean: Variability, Predictability and Change (CLIVAR) C20C+ Detection and Attribution Project (Stone et al. 2019)". The Hist ensemble is forced by historical anthropogenic and natural external forcing factors plus observational data of SST and sea ice (HadISST, Rayner et al., 2003). The Nat ensemble is forced by historical natural forcing factors and hypothetical "natural" SST and sea ice patterns where long-term anthropogenic signals were removed. Anthropogenic SST changes were estimated by taking the ensemble mean differences between the all-forcing historical runs and the natural-forcing historical runs of the CMIP5 AOGCMs. The multimodel averaged anthropogenic signal was subtracted from the HadISST data, and the Nat sea ice was estimated by using an empirical function that computes observed sea ice concentrations from surface temperature (Stone et al. 2019). Please note that both the Hist and Nat ensembles have 2015 El Niño components in the spatial patterns of SST, but the prescribed long-term warming anomalies in SST are different from each other. We performed 100 member runs of the 2006-2016 period for both Hist and Nat. Please see Shiogama et al. (2013; 2014) and Stone et al. (2019) for details regarding the experimental design.

We also analyse the 100 member ensembles of 11-year simulations with 1.5ºC and 2.0ºC warming relative to preindustrial levels. We performed those experiments as a contribution to the HAPPI project (Mitchell et al. 2016, 2017, 2018; Shiogama

et al. 2019). Since the ensemble-averaged global warming of the CMIP5 Representative Concentration Pathway 2.6 (RCP2.6) experiments is 1.55ºC, for the 1.5ºC runs, we used the RCP2.6 anthropogenic forcing agents (e.g., greenhouse gases) at 2095 and the ensemble mean 2091-2100 averaged SST anomalies of the RCP2.6 runs of the CMIP5 AOGCMs. The SST anomalies (Supplementary Fig. 2, top panel) are changes in the CMIP5 multimodel mean SST for each month, between the decadal average of 2091-2100 RCP2.6 and the decadal average of 2006-2015 RCP8.5. We added those SST anomalies to the 2006-2016 observed SST data of HadISST. To estimate the sea ice concentration, we applied a linear sea ice-SST relationship estimated from observations (Supplementary Figs. 3-4) (Mitchell et al., 2017). For the 2.0ºC runs, we used the weighted sum of RCP2.6 and RCP4.5 (0.41×RCP2.6 + 0.59×RCP4.5) of the well-mixed greenhouse gas concentrations in 2095 and the ensemble mean 2091-2100 averaged SST anomalies of the CMIP5 AOGCM ensembles (Supplementary Fig. 2, middle panel) because the weighted sum of the global mean temperature change values of the ensemble-averaged CMIP5 RCP2.6 and RCP4.5 runs is 2.0ºC. Please see Mitchell et al. (2017) for details regarding the experimental design. Notably, these future simulations have the same components as the 2015 El Niño event in terms of the spatial patterns of SST, but the prescribed long-term warming anomalies in SST have been added. Therefore, we can investigate drought events when events like the 2015 El Niño occur under 1.5ºC and 2.0ºC warmed climates relative to preindustrial levels.

Furthermore, we run the 100-member 3.0ºC ensemble (10-year simulations based on the 2006-2015 HadISST data) as an extension of the HAPPI project. Following the original HAPPI methodology, we add SST and sea ice concentration anomalies that represent additional warming in a 3°C warmer world compared to preindustrial values. The SST anomalies (Supplementary Fig. 2, bottom panel) are changes in the CMIP5 multimodel mean SST for the decadal average of 2006-2015 in RCP8.5 and the decadal average of 2091-2100 in a combined scenario of RCP4.5 and RCP8.5, i.e., 0.686×RCP4.5 + 0.314×RCP8.5 (Lo et al. 2019). The CMIP5 multimodel mean global mean temperature in 2091-2100 is approximately 3°C warmer than the 1861-1880 mean in this combined scenario; hence, this scenario describes 3°C global warming above preindustrial levels. For the sea ice concentration anomalies, we find the coefficients of this linear relationship from pre-existing 1.5°C and 2°C SST and sea ice anomalies. We apply this relationship to the 3°C SST anomalies to estimate the sea ice concentration anomalies, which are then added to the observed 2006-2015 data (see Mitchell et al., 2017). Supplementary Figs. 3-4 show the sea ice concentrations in both hemispheres in the 1.5°C, 2°C and 3°C experiments. The same weightings for RCP4.5 and RCP8.5 in the combined scenario equivalent to 3°C warming are also applied to greenhouse gas concentrations. This study is the first to report results from the HAPPI extension (i.e., the 3°C runs) using MIROC5.

To compute the normalized values of EA-averaged $\triangle$P and $\triangle\omega_{500}$ of the Hist, Nat, 1.5ºC, 2.0ºC and 3.0ºC runs, we subtract a long-term mean value of a given single member of Hist-long and divide anomalies by the standard deviation value of that Hist-long member. This normalization process enables us to produce 100×10=1000 samples of normalized $\triangle$P and $\triangle\omega_{500}$ data for each of the Hist, Nat, 1.5ºC, 2.0ºC and 3.0ºC ensembles.

## 4 Changes in precipitation, burned area and fire emissions of $CO_2$ and $PM_{2.5}$

The difference patterns of surface air temperature (≈ prescribed SST difference patterns over the ocean) in Hist-Nat, 1.5ºC-Nat, 2.0ºC-Nat and 3.0ºC-Nat have greater warming in the Niño 3.4 region than the tropical (30ºS-30ºN) ocean averaged values (Fig. 4). The relatively higher warming in the Niño 3.4 region accompanies downward motion anomalies in the EA region (Fig. 5a), enhancing negative precipitation anomalies when an El Niño occurs (Figs. 5b). Notably, the prescribed SST difference between the Niño 3.4 region and the tropical ocean mean is larger in the 1.5ºC runs than in the 2.0ºC runs. As a result, the amplitude of negative precipitation in the 1.5ºC runs is slightly greater than that in the 2.0ºC runs, as mentioned below, at least in these ensembles. It is not clear why the ensemble average of the CMIP5 RCP2.6 runs (i.e., the prescribed

SST anomalies of the 1.5 ℃ runs) has a larger SST difference between the Niño 3.4 region and the tropical ocean mean than that of the weighted sum of RCP2.6 and RCP4.5 (the 2.0 ℃ runs).

The 10 member ensembles of Lestari et al. (2014) were too small to estimate probabilities of droughts. Our large ensemble simulations enable us to estimate the probabilities of drought exceeding the observed value. Historical anthropogenic climate change has significantly increased the chance of $\triangle P$ being more negative than the observed value from 2% (1-4%) in Nat to

9% (6-14%) in Hist (Fig. 6a). Here, we use the cumulative histograms of $100 \times 10 = 1000$ samples of $\triangle P$ to estimate the probabilities of $\triangle P$. The values in parentheses indicate the 10-90% confidence interval estimated by applying the 1000-time resampling: we randomly resample $100 \times 10$ data from the original $100 \times 10$ samples of $\triangle P$ and compute the probabilities of drought exceeding the 2015 observed value; we repeat the random resampling process 1000-times and consider the 10%-tile and 90%-tile values of the 1000 estimates of probability as the 10-90% bounds. Even if the 1.5°C and 2.0°C goals of the Paris

Agreement are achieved (in the 1.5°C and 2.0°C runs), the chance of exceeding the observed value significantly increases from 9% (6-14%) in Hist to 82% (76-87%) and 67% (60-74%), respectively. In the current trajectory of 3.0°C warming (in the 3.0°C runs), the chance of exceeding the observed value becomes 93% (89-96%).

By combining the $\triangle P$ of MIROC5 (Fig. 6a) and the empirical relationships in Fig. 2, we assess the historical and future changes in burned area and fire emissions of $CO_2$ and $PM_{2.5}$ (Figs. 6b-d). We consider uncertainties by combining randomly

resampled $\triangle P$ and resampled regression factors of Eq. 1: (i) we compute the regression factors of Eq. 1 using randomly resampled data (the same as the process used to estimate the uncertainty ranges of the regression lines); (ii) we randomly resample $100 \times 10$ data from the original $100 \times 10$ samples of $\triangle P$; (iii) we use the regression factors of (i) and the $100 \times 10$ $\triangle P$ samples of (ii) to compute the 1000 estimates of fire or emissions and estimate the probability of exceeding the observed values; (iv) the processes of (i)-(iii) are repeated 1000-times; and (v) the 10%-tile and 90%-tile values of the 1000 estimates

of the probabilities of exceeding the observed values are considered to be the 10-90% bounds. Historical anthropogenic drying has increased the probability of exceeding the observed values of the burned area (from 5% (0-18%) to 23% (3-52%)), $CO_2$ emissions (from 5% (0-15%) to 23% (3-47%)), and $PM_{2.5}$ emissions (from 2% (0-5%) to 24% (3-49%)), but these changes are not statistically significant due to the large uncertainties. In the 1.5°C, 2.0°C and 3.0°C runs, the chances of exceeding the observed values significantly increase for the burned area (93% (66-99%), 81% (50-95%) and 98% (84-100%), respectively),

$CO_2$ emissions (92% (72-98%), 81% (55-93%) and 98% (86-100%), respectively), and $PM_{2.5}$ emissions (93% (70-98%), 81% (54-94%) and 98% (85-100%), respectively).

We contextualize the estimated fire $CO_2$ emissions within the future emissions scenarios. Although the above analyses focus on the year when the 2015-like El Niño events occurred, long-term mean fire $CO_2$ emissions are also important for mitigation policies. Here, we use simulated June-November mean precipitation anomalies of 11 years (2006-2016), instead of using only

the 2015 data, and the empirical function of Fig. 2b to estimate the cumulative probability function of fire $CO_2$ emissions in the EA area in the 2.0°C runs (Fig. 7). The fire $CO_2$ emissions of the 11-year period including both El Niño and non-El Niño years (Fig. 7) are much less than those in the year 2015 with the major El Niño (Fig. 6c) due to small fire $CO_2$ emissions in the non-El Niño years (Fig. 2). However, these fire $CO_2$ emissions can have substantial implications for mitigation policies. The vertical lines in Fig. 7 are the year 2100 land-use $CO_2$ emission scenarios including fire emissions for the East and South East

Asia regions except China and Japan in the five shared socioeconomic pathway (SSP) scenarios from the Asia-Pacific Integrated Model/Computable General Equilibrium (AIM/CGE) (Fujimori et al., 2012). AIM/CGE is one of the integrated assessment models (economic models) that produced the emissions data of SSP scenarios for the Coupled Model Intercomparison Project Phase 6 and the 6th assessment report of the Intergovernmental Panel on Climate Change (Riahi et al. 2017, Fujimori et al. 2017). Please note that the year 2100 land-use $CO_2$ emissions are not linearly related to the SSP numbers,

because the SSP numbers did not indicate radiative forcing levels. The chances of exceeding the emissions of SSP1, 2, 3, 4 and 5 are 77% (70-84%), 34% (28-39%), 13% (10-18%), 37% (31%-41%) and 77% (70-84%), respectively. Although these probability values highly depend on the SSP scenarios, the results are substantial in all the SSP scenarios. Because the $CO_2$

emissions in the AIM/CGE model include a wider area and other emission sources than the EA fire emissions of $CO_2$, this comparison is conservative. In the SSP simulations of AIM/CGE, fire $CO_2$ emissions are computed by using functions of land-cover changes, and climate change effects on fires are not considered. Therefore, it is suggested that implementing climate change effects on fire $CO_2$ emissions in integrated assessment models can significantly affect SSP land-use $CO_2$ emissions and studies of mitigation pathways, which in turn would be highly relevant to national and global climate policies. We suggest that additional fire $CO_2$ emissions due to climate change should be considered in possible CMIP7 activities.


**5 Conclusions**

By applying the probabilistic event attribution approach based on the MIROC5 AGCM ensembles, we suggested that historical anthropogenic warming significantly increased the chance of severe meteorological drought exceeding the 2015 observations in the EA area during the 2015 major El Niño year (from 2% (1-4%) in Nat to 9% (6-14%) in Hist). By performing and analysing the HAPPI (1.5°C and 2.0°C warming) and HAPPI extension (3.0°C warming) runs, we showed that the probabilities of drought exceeding the 2015 observations will largely increase (82% (76-87%), 67% (60-74%), and 93% (89-96%), respectively).

Drying trends tend to exacerbate fire activities. By combining these experiments and the empirical functions, we also implied that historical anthropogenic drying had tended to increase the chances of the burned area, $CO_2$ emissions and $PM_{2.5}$ emissions exceeding the 2015 observations, but those changes were not statistically significant. In contrast, if the 2.0°C goal is achieved, the chances of exceeding the observed values will substantially increase for the burned area (from 23% (3-52%) in Hist to 81% (50-95%) for 2.0°C), $CO_2$ emissions (from 23% (3-47%) to 81% (55-93%)) and $PM_{2.5}$ emissions (from 24% (3-49%) to 81% (54-94%)). These results agree well with Lestari et al. (2014) and Yin et al. (2016) who showed that the AOGCM ensemble of CMIP5 projected future long-term trends of drying and enhanced fire carbon emissions. We further suggest that the risks of drought and fire significantly increase when events like the 2015 El Niño occur in future warmer climates even if the 1.5°C and 2.0°C goals are achieved. The impacts of these changes on droughts, burned area and fire emissions should be reduced by adaptation investments.

If we cannot limit global warming to 2.0°C and it reaches 3.0°C as expected from the current "emissions gap" (United Nations Environment Programme 2018), the chances of exceeding the observed values further increase for the burned area, $CO_2$ emissions and $PM_{2.5}$ emissions. Although the differences between 2.0°C and 3.0°C are not statistically significant for the burned area and the $CO_2$ and $PM_{2.5}$ emissions, the 50th percentile values of probabilities exceeding the 2015 observations first reach approximately 100% in the 3.0°C runs. These additional changes relative to 2.0°C indicate the effects of the failures of mitigation policies. Conversely, these changes indicate the potential benefits of limiting the current trajectory of 3°C global warming to the Paris Agreement goals.

Forest-based climate mitigation has a key role in meeting the goals of the Paris Agreement (Grassi et al., 2017). We also suggested that changes in fire $CO_2$ emissions due to future warming can increase the need for modifying fire $CO_2$ emission scenarios for future climate projections. Although we focused on the influences of climate change on burned area and fire emissions, land use and land cover changes are also important factors. To avoid fire intensification due to drying climates, effective land management policies for protecting forests and peatlands are necessary (Marlier et al., 2015; Kim et al., 2015; Koplitz et al., 2016; World Bank 2016).

This study is based on the single model ensembles using the particular SST anomaly patterns. A future work to compare multi model simulations using multiple estimates of warming patterns in SST would be useful.

**Data availability.**

The data of MIROC5 model, ERA-I, GPCP and GFED4s used in this article can be download from
https://portal.nersc.gov/c20c/ , https://www.ecmwf.int/en/forecasts/datasets/reanalysis-datasets/era-interim ,
https://www.esrl.noaa.gov/psd/data/gridded/data.gpcp.html , and https://www.globalfiredata.org/data.html , respectively. The
data of AIM/CGE can be accessed by contacting the corresponding author.

295 .

**Author contributions.**

HS, RH, TH, SF and SC designed the analysis. HS performed the analysis and wrote the first draft of the paper. HS, YTEL
and DM proposed and performed the HAPPI extension runs. All authors contributed to the interpretation of the results and to
the writing of the paper.

**Competing interests.**

The authors declare that they have no conflict of interest.

**Acknowledgements**

We thank the reviewers and the editor for their useful comments. The MIROC5 simulations were performed using the Earth
Simulator at JAMSTEC and the NEC SX at NIES.

**Financial support.**

This study was supported by ERTDF 2-1702 (Environmental Restoration and Conservation Agency, Japan), the Integrated
Research Program for Advancing Climate Models (TOUGOU, JPMXD0717935457) and the Climate Change Adaptation
Research Program of NIES. This research used the science gateway resources of the National Energy Research Scientific
Computing Center, a DOE Office of Science User Facility supported by the Office of Science of the U.S. Department of
Energy under contract no. DE-AC02-05CH11231.

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

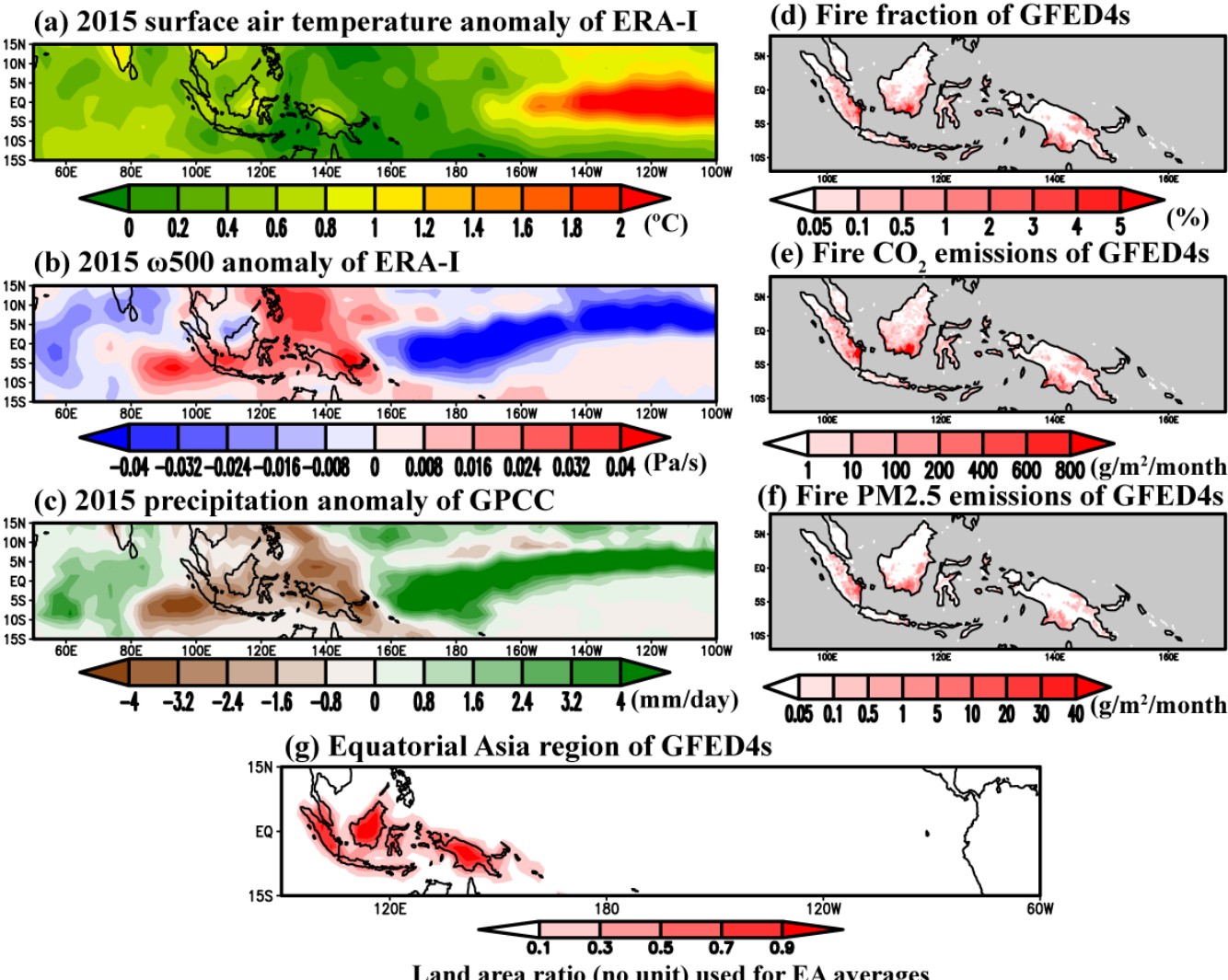

**Figure 1: The observed climate conditions and fires.** The June-November 2015 averaged anomalies of (a) surface air temperature (°C) and (b) vertical pressure velocity at the 500-hPa level (Pas$^{-1}$, downward motions are positive) from ERA Interim reanalysis data (Dee et al. 2011) relative to the 1979-2016 mean. (c) The June-November 2015 averaged anomalies of precipitation from GPCP (Adler et al. 2003) (mm/day). The right panels indicate (d) fire fraction (%), (e) fire $CO_2$ emissions (gm$^{-2}$ month$^{-1}$) and (f) fire $PM_{2.5}$ emissions from GFED4s (van der Werf et al. 2017) during June-November 2015. (g) The red area indicates the EA region of the GFED4s. We use this definition of the EA area. Shading shows the land area ratio (no unit) used for weighting in the computation of EA averages.

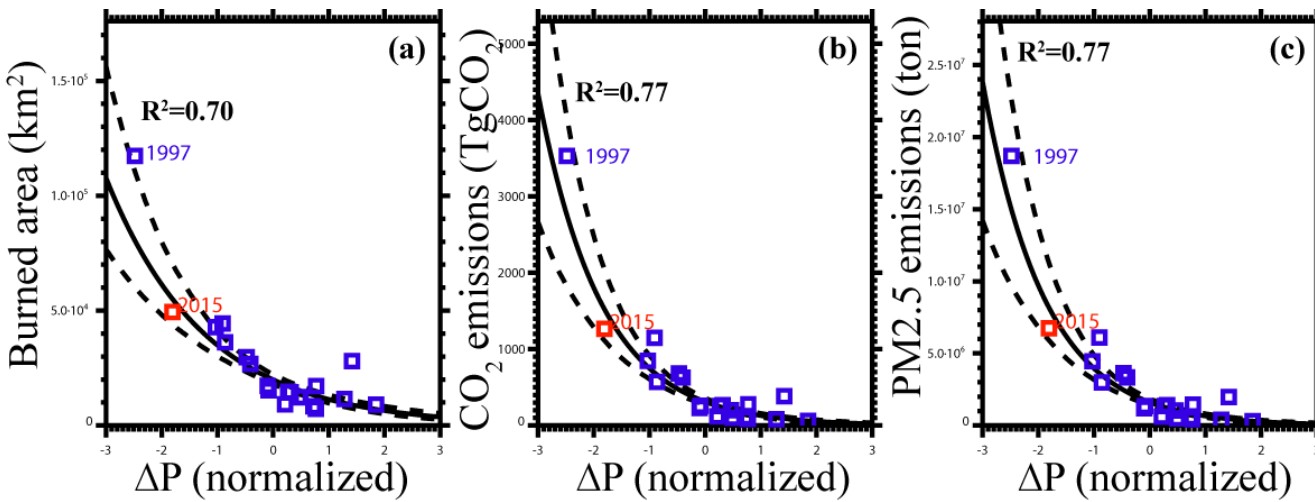

**Figure 2: Empirical relationships between observed precipitation anomalies, burned area and fire emissions in the EA area during 1997-2016.** The horizontal axes are the normalized June-November mean precipitation anomalies (no unit) of the GPCP. The vertical axes denote (a) burned area ($km^2$), (b) $CO_2$ emissions ($TgCO_2$) and (c) $PM_{2.5}$ emissions (ton) of GFED4s. The year 2015 values are indicated by red squares. Solid and dashed lines indicate the best estimates and 10-90% confidence intervals of the fitting curves from Eq. 1, respectively.

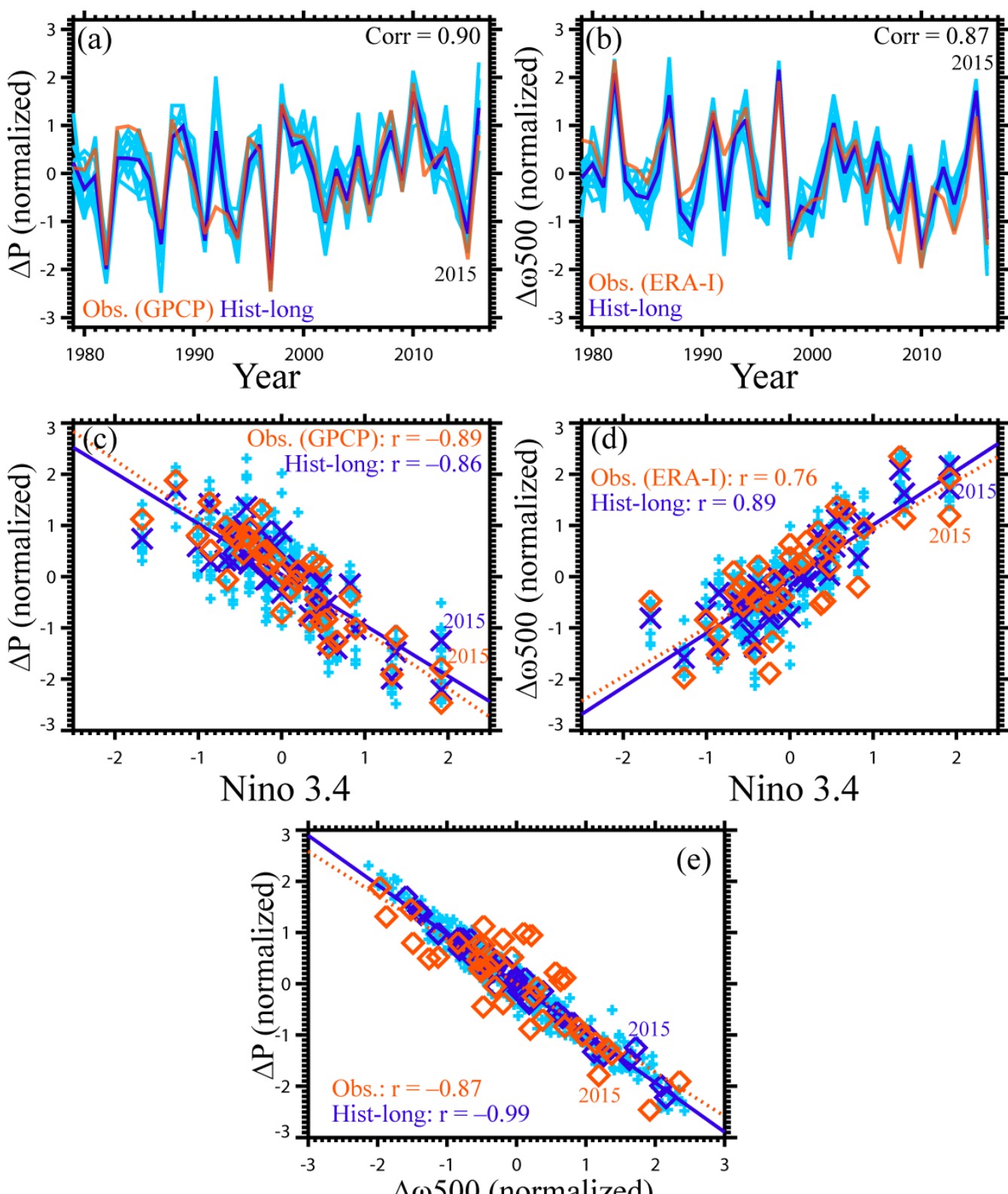

**Figure 3: Evaluations of the MIROC5 simulations of the EA averaged precipitation and vertical air motions. Top panels show the normalized June-November mean time series of (a) $\Delta P$ (no unit) and (b) $\Delta \omega_{500}$ (no unit). Red lines are the observations. Light blue lines are the 10 ensemble members of Hist-long, and blue lines are the ensemble mean. The other panels are scatter plots of (c) $\Delta P$ and the Nino 3.4 index (°C), (d) $\Delta \omega_{500}$ and the Nino 3.4 index and (e) $\Delta P$ and $\Delta \omega_{500}$. Red diamonds are the observed values. Small light blue crosses are the 10 ensemble members of Hist-long, and large blue diamonds indicate the ensemble mean values. The red**
**and blue lines indicate the regression lines of the observations and the ensemble averages of Hist-long, respectively.**

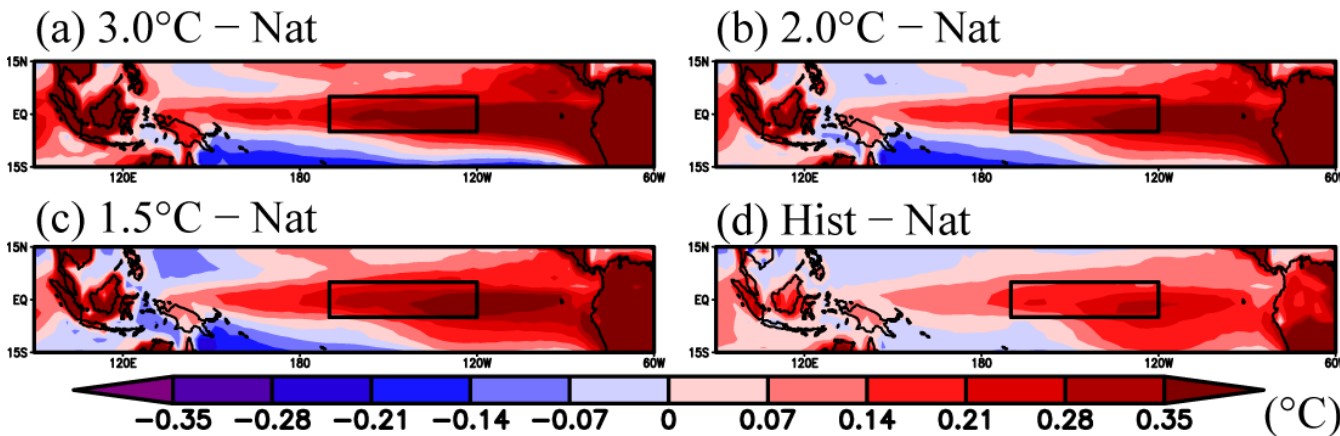

**Figure 4: Surface air temperature warming patterns in 2015. (a) △T differences between 3.0 ºC and Nat (ºC). The 30ºS-30ºN ocean averaged value is subtracted. The black box indicates the Nino 3.4 region. The other panels are the same as panel (a) but for (b) 2.0 ºC minus Nat, (c) 1.5 ºC minus Nat and (d) Hist minus Nat.**


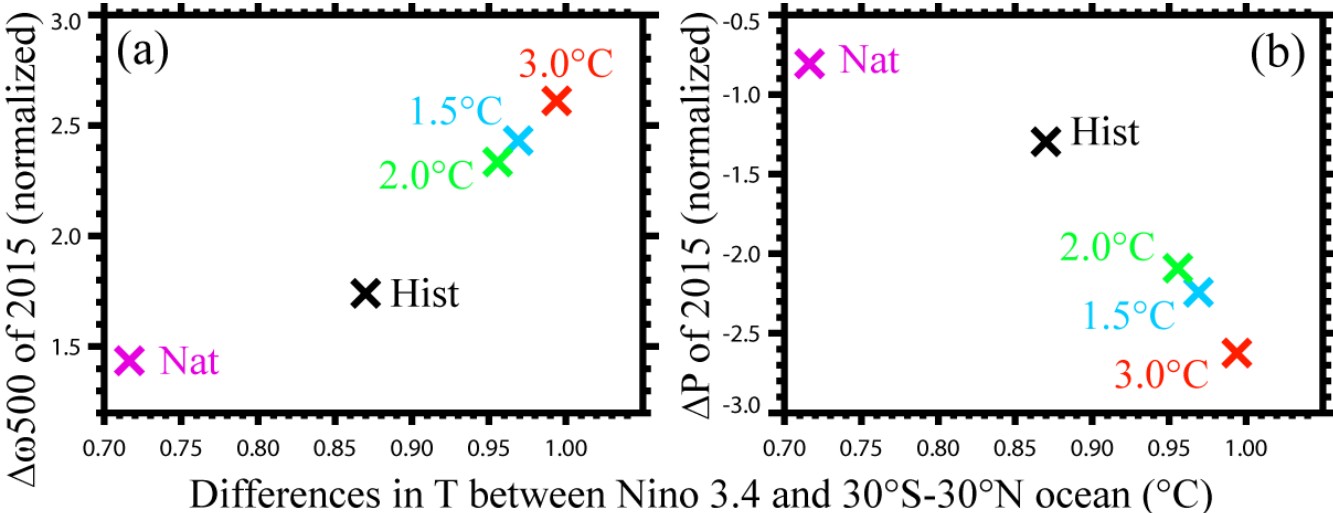

**Figure 5: Relationships between Niño 3.4 warming and EA vertical motion and precipitation anomalies of the ensemble mean. The horizonal axes show differences in the 2015 T anomalies between the Niño 3.4 area and the 30ºS-30ºN ocean (ºC). The vertical axes are (a) $\triangle\omega_{500}$ (no unit) and (b) $\triangle P$ (no unit) for the year 2015. Crosses denote the ensemble averages of Nat (purple), Hist (black), 1.5ºC (light blue), 2.0ºC (green) and 3.0ºC (red).**


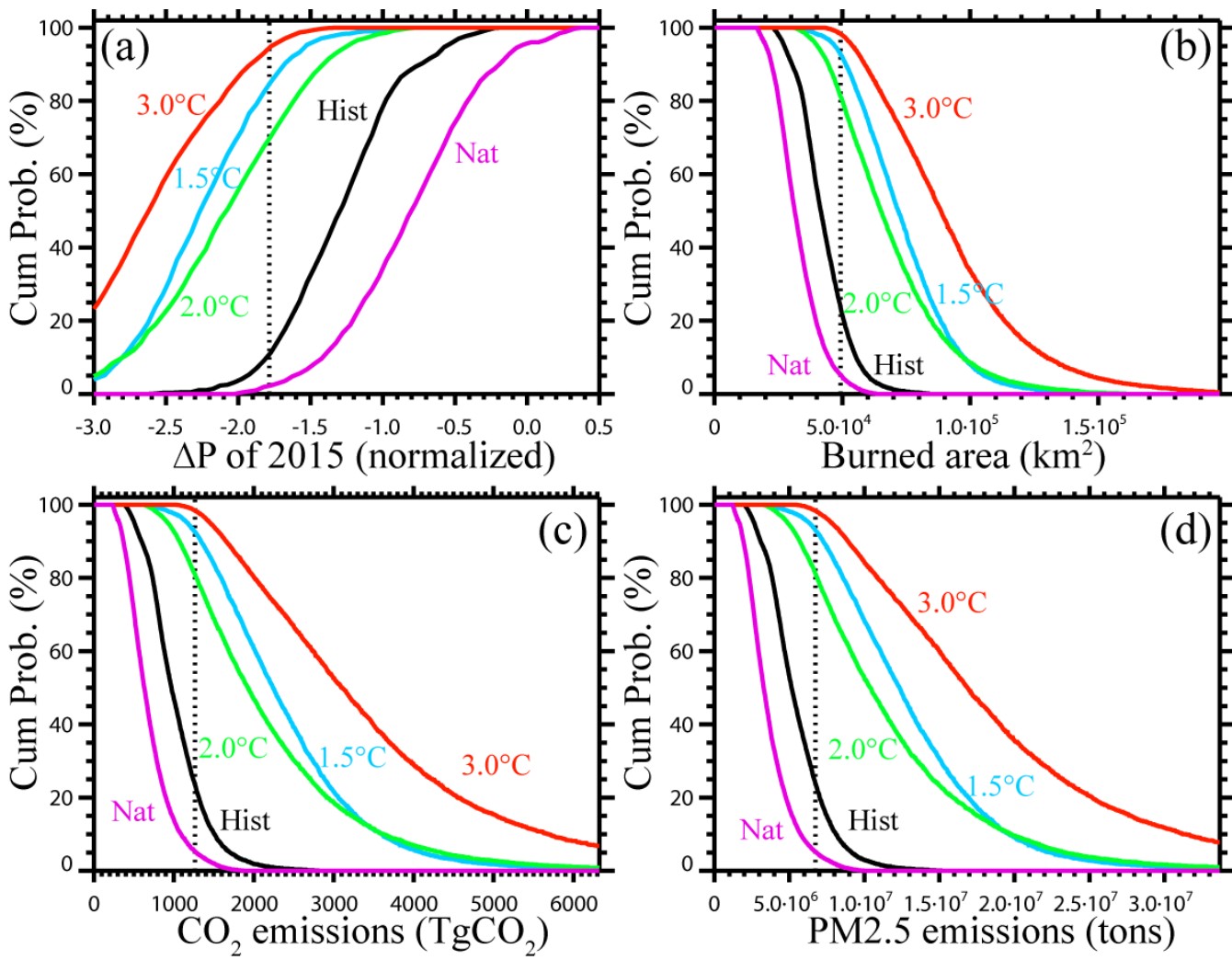


**Figure 6. Changes in the cumulative probability functions. (a) The vertical axis indicates the probability (%) of ΔP being lower than a given horizontal value (no unit). Solid lines denote the 50% values of the 1000 random samples of the Nat (purple), Hist (black), 1.5ºC (light blue), 2.0ºC (green) and 3.0ºC (red) ensembles. The vertical dotted line is the observed 2015 value. The other panels show the probabilities of exceeding the given horizontal values for (b) the burned area (km$^2$), (c) CO$_2$ emissions (TgCO$_2$)**

**and (d) PM$_{2.5}$ emissions (tons).**

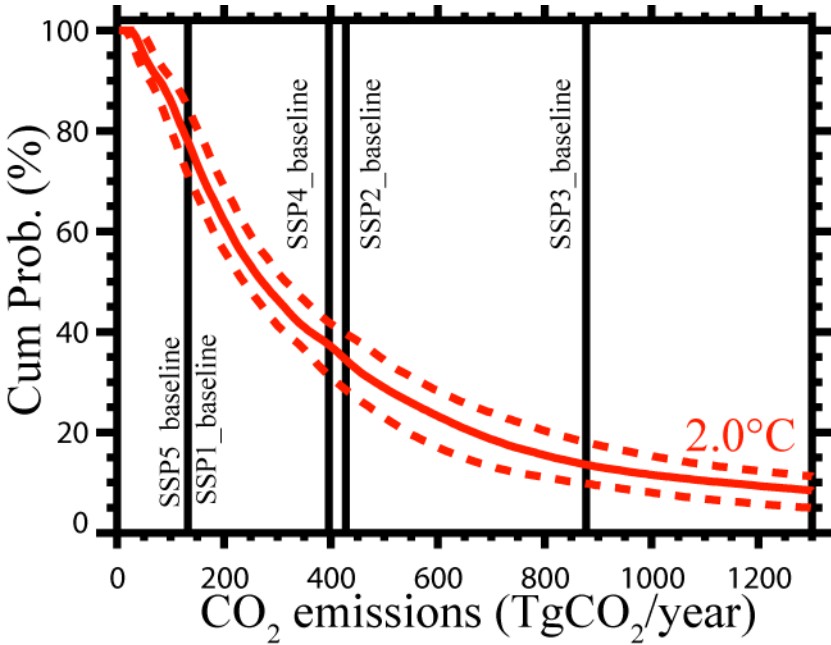

**Figure 7: The red curves are the cumulative probability function of $CO_2$ emissions (TgCO$_2$/year) during June-November of 2006-2016 for the 2.0ºC runs. Solid and dashed lines denote the 50% values and the 10-90% confidence intervals, respectively. The vertical lines indicate the year 2100 annual land-use $CO_2$ emission scenarios (including fire emissions of $CO_2$) for the East and South East Asia regions, except China and Japan for the 5 SSP baseline scenarios of the AIM/CGE model.**