# Peer review of "Historical and future anthropogenic warming effects on droughts, fires and fire emissions of CO2 and PM2.5 in equatorial Asia when 2015-like El Niño events occur"

_Earth System Dynamics, 2019_

## Referee Comment (RC1) · Anonymous Referee #1 · 10 Sep 2019

The title of the study is confusing. Is it about historical and future anthropogenic warming effects on the year 2015? — could future anthropogenic warming have an impact on a past year? Equally confusing is the abstract. For instance, "we suggest that historical anthropogenic warming increased the chances of meteorological droughts exceeding the 2015 observations in the EA area . . ..." (line 15-29). What does it mean exactly? Which period are those claims referring to?

The abstract lacks fundamental clarity, so does the paper. It seems to me that the authors have not sorted out a coherent logic chain to tell a concrete story. Instead, the paper presents a series of model results without a clear rationale to make sense of

it. Last but not least, several figures of this paper have been documented by previous studies as cited in the paper (in slight variations); I do not see added value from the duplication. Therefore, I recommend rejecting this paper in its current form.

Please see a few technical issues below (not an exhaustive list):

Line 47: "this study has three aims." What is its relevance given the studies mentioned in the previous paragraph?

Line 48: A higher-level introduction of the "probabilistic event attribution approach" is necessary for the readers to understand the concept. While technical details could refer to published papers, the general principle should be properly introduced.

Line 55-56: On what temporal horizon? This point is not clear at all. After reading the entire paper, it appears to me that the warming scenarios (1.5, 2.0, or 3.0 C) is defined regarding the reference year 2100. However, the simulated results in 2015 are discussed. This setting is problematic since the emission changes in each scenario are not linear across the century, what does it mean when comparing the first few years? The starting and ending point of the simulations are not clearly stated. The use of those simulations is not adequately justified.

Line 64: It will be helpful to state clearly how many ensemble members for each scenario, covering which period, based on what emission trajectory.

Line 77: "Although socio-economic factors are important for fire activities, we only examine the effects of climate change in this study." "climate change" here refers to simulated climate given different forcing scenarios, which by definition accounts for socio-economic factors. Maybe, the authors are referring to local land-use change impact? It is important to make those distinctions to make sense of the results.

Section 2. Compared to the papers cited here, I do not see any new contribution from this section. It just shows what has been done, without any new data or insight.

Line 118: "Please note that both the Hist and Nat ensembles have the same spatial

SST patterns as the 2015 El NinÌČo event". How so?

Line 124: If the simulation periods are 2006-2015, how come are they future simulations???

Line 136: Why is showing sea ice (Fig. 5) relevant to this study? Similarly, it is not clear about the role of results showing in Figure 4 to Figure 7.

Line 169: Are you comparing fire and fossil fuel emissions from Japan? Not clear.

Line 192-194: By now, it is still difficult to understand what do they mean.

Line 197: "somewhat"?

Line 192-212: The conclusion is far reached.

---

## Referee Comment (RC2) · Anonymous Referee #2 · 4 Oct 2019

General comments:

The authors present a policy-relevant study of changes in precipitation deficits and fire metrics in equatorial Asia (EA) during a 2015-like El Niño event and in a decadal-average sense at 1.5°, 2°, and 3°C warming levels. Results are based on a factual-counterfactual probabilistic event attribution approach in a MIROC5 AGCM large ensemble framework. The following questions are explored:

- Did historical climate change increase the probability of the 2015 event?

- How will probabilities of drought, fire and fire emissions change when a major El

[Figure]

Niño event similar to the one in 2015 occurs in 1.5°C, 2.0°C, 3.0°C climate.

Authors find that historical anthropogenic forcing has increased the likelihood of a drier-than-2015 El Niño-driven precipitation deficit in the EA from 2% to 9%. At 1.5°C of warming, a drier event is 82% likely to occur. At 2°C of warming, the probability of a drier event drops to 67% likely (for reasons that are not entirely clear) but increases to 93% at 3°C of warming. This increased risk of drier conditions during El Niño events has ramifications for burned area extent, CO2 and particulate emissions in the regions.

The paper could be a valuable addition to the current body of literature on extreme events in warmer climates and the figures presented are clear and easy to interpret. There is care taken to connect findings to policy considerations whenever possible, particularly possible underestimates of EA $CO_2$ emissions under climate change scenarios. However, the interesting findings would benefit from additional detail on the method, model experimental framework, relationships between relevant processes, and, most critically, on the value-add gained by using a 100-member large ensemble as opposed to 10-member ensembles. Addressing the specific comments below should more than adequately clarify and strengthen conclusions.

Specific comments:

I recommend omitting the phrase: "the year" from the title.

Abstract On what is the statement "caused" based on? [L14] What is the dry season in Equatorial Asia (i.e., in terms of months)? [L15] The acronym PM2.5 is not yet defined [L21]

Introduction It would be helpful to explicitly define when EA dry season occurs. . . Is the EA region the same as the SEA SREX region? If not, the specific latitude and longitude boundaries should be given. [L32-34]

Could you explain the ENSO phenomenon and how and why it "enhanced severe drought" to help guide readers? How did the 2015 drought compare to other ENSO

events? This can be done through an explanation of the relationship between Walker circulation and convection and by including relevant citations on how the 2015 event compares to other El Nino in terms of effect on EA climate. [L32-34]

Is the whole EA region considered tropical peatland? Can that qualification be defined (i.e. what is tropical peatland) and can it be explained why the region is susceptible to biomass burning? [L33]

Can you elaborate more on the findings of Lestari et al. 2014? It will help readers understand how this new study extends the findings. [L43]

Question 1: Did historical climate change increase the probability of the 2015 event? How is historical climate change defined? Is it with respect to a certain base period? How is "change" defined in the presence of natural variability? [L47]

Can you elaborate on the probabilistic event attribution approach used in this study? It will help readers who are not familiar with detection and attribution techniques understand the opportunities and limitations of these approaches. How were they used in these cited papers? What were some of the key findings? [L48-54]

I also recommend the following edit: "We use a probabilistic event attribution approach similar to Lestari et al. (2014), but our results are based on 100-member large ensembles of the MIROC5 AGCM with and without anthropogenic warming as opposed to 10-member ensembles." Then, a statement should be made about why the large ensembles were necessary and important to use. Justifying and highlighting the importance of large ensembles is a key point, especially for a special journal edition dedicated to large ensembles. [L48-50]

Question 2/3: Could these two sections be combined into one section about risk at 1.5, 2.0, and 3.0°C warming? [L55]

Just a small comment but can the connection between the initiation of the HAPPI project and the Paris agreement be smoothed out a little? Was the HAPPI project

initiated in response to or to inform the Paris agreement? [L56-61]

In regards to "Although socio-economic factors (e.g., conversions of forest and peat-lands to agriculture and plantations of oil palm) are also important for fire activities (Marlier et al., 2013, 2015; Kim et al., 2015), we only examine the effects of climate change in this study." Does this mean land-use change is not considered? What are the relevant "effects of climate change" on these events (i.e. warmer mean temperature, circulation changes, changes in ENSO?)

Empirical functions: Can you elaborate on your observational dataset choices? Why did you choose the reanalysis products you use? How are the enhanced fire fraction, fire CO2 emissions and fire $PM_{2.5}$ emissions computed in the Global Fire Emissions Database?

Could a figure demonstrating the relationship between burned area, CO2 emissions, and particulate emissions be included? Is there a linear relationship between burned area and emissions?

What are the empirical functions used for in this study?

Model simulations: Can you provide a further description of the MIROC5 AGCM? I.e. what is the horizontal resolution of the atmosphere? What observed SSTs specifically were used, particularly for the "natural" SST? How was the "long-term anthropogenic signal" defined and removed? Most importantly, what fire model is used? How is it related to the land surface state and coupled to the atmosphere? What triggers a fire in the model? How are CO2 and $PM_{2.5}$ concentrations determined for a given event? [L103-121]

What are the "corresponding standard deviation values"? [L108]

Throughout the study, the descriptions of the figures are a little brief. In this case: "The precipitation and vertical motion anomalies are closely related to the Nino 3.4 SST (an index of El Niño Southern Oscillation) in the observations, and the MIROC5 model

represents these relationships well (Figs. 3c-e)." How are they related (i.e., subsidence and reductions in rainfall during an El Niño)? What does "represent these relationships well" mean (i.e., significantly correlated with observations)? [L108-110]

How were "prescribed long-term warming anomalies in SST" defined? From Figure 4, I can see that there are spatial differences in warming, where do they come from? These details are likely important to the overall interpretation of the results and it would benefit the reader not to have to search for methodological descriptions in other studies or elsewhere in the paper. [L125-126]

The colorbar seems to be saturated in the bottom panel of Figure 4 over much of the Northern hemisphere, could the scale be adjusted to accommodate the 3°C mean difference? Is there a difference between the respective Figure 5 top and middle panels? Could you detail how these results were reached using the cumulative density functions? Particularly, how did the use of large ensembles affect the results? Does the "chance of exceedance" change with fewer members? [L154-158]

Can you comment on why the chance of precipitation reduction exceedance more probable in the 1.5° scenario than the 2° scenario? [L157]

"2015 CO2 emission of Japan due to fossil fuel consumptions" is a missing a citation [L168-169]

I am sorry I may have missed something, but what is the AIM/CGE model used for? Was it introduced in the methods section? [L179-180]

---

## Referee Comment (RC3) · Anonymous Referee #3 · 14 Oct 2019

General comments: This study examines global warming impacts on fire activities, focusing on the burned area and fire emissions of CO2 and PM2.5 over equatorial Asia. Considering June-November 2015 when a strong El-Nino induced a large decrease in precipitation over the area, the authors examine changes in the probabilities of droughts and fire activities due to anthropogenic influences using the MIROC5 AGCM large ensemble (100 members) simulations. They find increased probabilities of the droughts and fire activities as global warming become stronger. In particular, they show that 3.0 degree warming that represents the current mitigation policies would bring severe droughts and increased fire activities due to the intensified El Nino at near 100% chance. I find this paper overall well written, providing interesting and policyrelevant results. However, there are a few issues, mostly related to uncertainty factors, which need to be improved through revision.

Major points: 1. Model dependency: It would be useful to discuss limitations of the atmospheric model experiments and its possible impacts on the results. Particularly, precipitation changes in the future warming simulations are shown to be critical for determining changes in fire burned area and $CO_2$ and PM2.5 emissions (Fig. 8), but atmospheric models tend to have large biases in precipitation over the Tropics partly related to the omission of air-sea coupling. It seems that normalized precipitation is used to overcome this problem but some justification would be needed with showing precipitation bias of the model. In addition, future projections of precipitation look highly dependent on the SST change patterns (Fig. 4). Uncertainty in these SST change patterns needs to be discussed as well.

2. New findings: New results compared to previous studies are not clearly explained, in particular, in view of Lestari et al. (2014). What advances have been achieved by increasing ensemble size? Adding more information on this would be helpful such as how to construct ensembles and how uncertainty is assessed with the large ensemble simulations. Also, the empirical relation between precipitation and fire activities is used to estimate future changes in fire activities and the authors consider its uncertainty somehow in their analysis. I think this part is important and more details needs be provided on its uncertainty ranges and associated impacts on main results. See my specific points below.

3. Implications: The last part on implications is rather confusing and hard to follow. I would suggest rephrasing it for better understanding. For example, it is unclear what are exactly compared between MIROC5-based estimations and diverse SSP scenarios: fire $CO_2$ emissions due to climate change versus land use $CO_2$ emissions? From this comparison, the authors seem to suggest that additional fire $CO_2$ emissions due to climate change should be considered in SSP scenarios, but this interpretation is not that clear at the present form. I am wondering if it can be made more specific by
suggesting how much increase in CO2 emissions should be added, for example.

Specific points: Title: "year 2015" sounds a bit strange to be connected with "future" warming effects. How about saying "2015-like" or similar instead.

L87-88: How is the EA box selected? I think it can be adjusted (e.g., narrower in zonal direction) to better capture the P decrease area. Or it doesn't matter since only land is considered? Please clarify this.

L91-92: In line with "precipitation anomalies and accumulated water deficits", wouldn't it be better to use accumulated precipitation like SPI?

L93, L108: "divided by standard deviation". Can we assume normality for precipitation and omega anomalies? Area averaged 6-month mean values might be okay but a quick check would be useful.

L115: I would suggest providing more details on how "long-term anthropogenic signals were removed" as SST patterns are important for determining precipitation responses to El Nino.

L122: "100 member ensembles during 2006-2015 with 1.5 degree and 2.0 degree warming". Do it mean that ensemble runs are performed only for Plus15 and Plus20 or there are 100-member HIST runs for 2006-2015 as well?

L141-144: This way of sampling looks important to capture uncertainty arising from internal variability, and showing resulting spreads in P and omega responses in Fig. 7 would be interesting. Also, it would be useful to explain here how to construct CDF using 1000 samples and estimate probabilities exceeding the observed value and its 10-90% confidence intervals.

L150-153: Why stronger El Nino (and P responses) are simulated in 1.5 degree warming simulations than 2.0 degree ones? Some discussion needs to be provided. Does it occur in all 1000 samples? Do other HAPPI models share this or is this a characteristic of MIROC5?
L161: "1000 random samples of the regression factors in Eq. 1". Please provide details given its importance. Also see my major comment.

L163-164: Please explain how to assess significance of this change.

L169: Why is the emission of Japan used as reference here?

L172: First sentence. This needs to be mentioned clearly above and also in figure captions to avoid confusing.

L172-188: This paragraph and Fig. 9 are hard to follow with many skips and limited explanations. Please consider rephrasing it. See my major comment above.

L196: "82%, 68%, and 93%". Please add uncertainty ranges or indicate these are ensemble means or medians. Same for L204.

L199-202: Model dependency issue is here. How representative is MIROC5 projected precipitation in the future? Any comparison with other models would be useful. See my major comment above.

L205: "additional changes". Are these significant?

L209: "modifying fire CO2 emissions scenarios". Can authors suggest how much modification is needed? See my major comment above.

Fig. 1: Line 342: "left panels" should be "right panels".

Fig. 2: There seems to be a stronger case than 2015, perhaps 1998? Where is 1982 that has also a stronger P decrease in Fig. 3? It may affect fitted curves.

Fig. 3: Indicating 2015 case in time series and scatter plots would be useful. Is there any underestimation or overestimation by models in P and omega responses?

Fig. 6: It's not clear why difference from NAT is shown even for future changes. Is this for 2015 or using all years?

Fig. 7: Is this also for 2015? Related to my major comment on model dependency

issue, are these are supported by other coupled models?

Fig. 9: Difficult to understand. How is the CDF of $CO_2$ emissions (red curves) estimated? Are these $CO_2$ emissions only due to increased fire over equatorial Asia?

---

## Author Response (AR1)

**Dear Prof. Nicola Maher,**

We are resubmitting the revised manuscript of esd-2019-46 by Hideo Shiogama et al. Based on the advice of the reviewers, we changed the title to "Historical and future anthropogenic warming effects on droughts, fires and fire emissions of CO2 and PM2.5 in equatorial Asia when 2015-like El Niño events occur".

>Thank you for the submission of this interesting manuscript to this special issue.
>While the reviews were generally positive, I am asking for major revisions to your manuscript.
>In particular the reviewers raised concerns about the context of your study in relation to previous
> work, particularly Lestari et al. (2014).

We explain the probabilistic event attribution approach, the difference between Lestari et al. (2014) and the present study, and why 100 member ensembles are necessary in lines 54-95 and 263-266. Because the 10 member ensembles of Lestari et al. (2014) are too small to examine how historical climate changes affected the probability of extreme events, we use 100 member ensembles [lines 74-76].

>There was also some confusion around the implications section, abstract and presentation
> which require revision to make the manuscript clearer and the results more understandable.
> I look forward to reading the revised manuscript.

Because our descriptions of experimental designs were not enough in the original manuscript, the reviewers were confused. Therefore we have added many sentences to explain our experiments and logics.

To help the reviewers, we have highlighted the changes in red colours in the clean version of the manuscript. We hope the revised manuscript satisfies you and the reviewers.

Best regards, Hideo Shiogama

**Reply to reviewer #1**

Thank you very much for your helpful comments. Based on your comments, we have improved the manuscript.

The title of the study is confusing. Is it about historical and future anthropogenic warming effects on the year 2015? — could future anthropogenic warming have an impact on a past year?

Based on the advice of the other reviewers, we changed the title to "Historical and future anthropogenic warming effects on droughts, fires and fire emissions of CO2 and PM2.5 in equatorial Asia when 2015-like El Niño events occur". We investigated the historical anthropogenic warming effects on the 2015 event and also assessed how future warming can affect droughts, fire and emissions when 2015-like El Niño events occur in 1.5, 2.0 and 3.0 degree Celsius warmed climates. [lines 16-22, 59-95 and section 3].

Equally confusing is the abstract. For instance, "we suggest that historical anthropogenic warming increased the chances of meteorological droughts exceeding the 2015 observations in the EA area : : ..." (line 15-29). What does it mean exactly? Which period are those claims referring to?

Please note that we compare factual condition simulations (Hist) and counterfactual natural forcing condition simulations (Nat). In both ensembles, the SST is prescribed as that the 2015-like El Niño occurs. In the Nat ensembles, anthropogenic warming from the preindustrial to the present is removed from the SST data. By comparing these two large-member ensembles, we suggest that historical anthropogenic warming increased the probability of meteorological droughts exceeding a given threshold when the 2015 El Niño event occurred. [lines 16-18, 59-78, 146-159]

The abstract lacks fundamental clarity, so does the paper. It seems to me that the authors have not sorted out a coherent logic chain to tell a concrete story. Instead, the paper presents a series of model results without a clear rationale to make sense of it. Last but not least, several figures of this paper have been documented by previous studies as cited in the paper (in slight variations); I do not see added value from the duplication. Therefore, I recommend rejecting this paper in its current form. Our explanations were not enough in the original manuscript. Therefore we have added many explanations of the experimental designs and our logics as mentioned below. We hope that these explanations help the reviewers and readers to better understand our results.

For example, in lines 59-78, we explain the probabilistic event attribution approach as follows: "Although Lestari et al. (2014) showed the anthropogenic effects on the historical trends in droughts, it is not clear how historical climate changes affected the particular drought event of 2015. Because extreme events can occur by natural variability alone, it is difficult in principle to attribute a particular event to anthropogenic climate change. However, comparisons of observations and large ensemble simulations can help us evaluate the degree to which human influence has affected the probability of a particular event (Allen 2003). Such an approach is called probabilistic event attribution (PEA) (Pall et al. 2011, Shiogama et al. 2013). In the PEA approach, two large ensemble simulations (e.g., 100 members) are generally performed. The first is historical simulations of an AGCM driven by the historical values of anthropogenic (e.g., greenhouse gases) and natural forcing (solar and volcanic activities) agents and by the observed sea surface temperature (SST) and sea ice concentration (SIC). The second is counterfactual natural runs driven by preindustrial anthropogenic and historical natural forcing agents and by the observed values of SST and SIC cooled according to estimates of anthropogenic warming (Stone et al. 2019) (see section 3 for more details). Note that the components of interannual variations in the SST data are not modified in the natural forcing ensemble. Therefore, for example, we can assess how anthropogenic warming affected the probabilities of drought events exceeding the observed value in the 2015 major El Niño year by comparing the distributions of members in historical and natural forcing ensembles. In this study, based on the PEA approach, we examine whether historical climate changes increased not only the probabilities of drought but also those of fire and fire emissions of CO2 and PM2.5 during the June-November dry season of 2015. Because the 10 member ensembles of Lestari et al. (2014) are too small to estimate probabilities of extreme events, we use 100 member ensembles of Shiogama et al. (2014). The lower computing costs of AGCM than AOGCM enable us to perform large ensembles, which are necessary for PEA."

In lines 79-95 we explain our future experiments and their relationships to the Paris Agreement goals and the emission gaps as follows: "Although Lestari et al. (2014) and Yin et al. (2016) showed increases in droughts and fires in the future projection ensembles of AOGCMs, it is not clear how future anthropogenic warming affects droughts and fire when events like the 2015 El Niño occur in a future warmer climate. It is important to investigate changes in extreme events at 1.5°C and 2.0°C warming levels to inform stakeholders after that the Paris Agreement set the 2°C long-term climate stabilization goal and moreover state pursuing 1.5 °C for stabilization (United

Nations Framework Convention on Climate Change 2015), but Lestari et al. (2014) and Yin et al. (2016) did not perform such analyses. In this study, we examine how the probabilities of drought, fire and fire emissions of CO2 and PM2.5 would change when major El Niño events like 2015 occur under 1.5°C and 2.0°C warmed climates. We analyse large (100-member) ensembles of the MIROC5 AGCM under the Half a degree Additional warming, Prognosis and Projected Impacts (HAPPI) project, which was initiated in response to the Paris agreement (Mitchell et al., 2016, 2017, 2018; Shiogama et al., 2019). These MIROC5 HAPPI ensembles have been used, for example, to study the changes in extreme hot days (Wehner et al., 2018), extreme heat-related mortality (Mitchell et al., 2018), tropical rainy season length (Saeed et al., 2018) and global drought (Liu et al., 2018) at 1.5°C and 2.0°C global warming. There is a significant "emissions gap", which is the gap between where we are likely to be and where we need to be (United Nations Environment Programme 2018). The current mitigation policies of nations would lead to global warming of approximately 3.2°C (with a range of 2.9-3.4°C) by 2100 (United Nations Environment Programme 2018). Therefore, it is worthwhile to compare changes in extreme events and impacts in cases where the 1.5°C and 2.0°C goals are achieved or not. Therefore, we perform and analyse a large ensemble of a 3.0°C warmed climate."

It seems that you believe that Fig. 2 is a duplication. We do not insist to say Fig. 2 is a new result [lines 126-128]. Our main aims are that we combine those empirical functions (not new) with the large ensemble simulations of droughts to assess the climate change effects on fire and emissions when 2015-like El Nino events occur at the warming levels of the present, the Paris agreement goals and the current mitigation trajectory (new results).

Please see a few technical issues below (not an exhaustive list): Line 47: "this study has three aims." What is its relevance given the studies mentioned in the previous paragraph?

We explain the relationships between the previous studies (Lestari et al. 2014 and Yin et al. 2016) and this study in lines 59-95 and 263-266.

Line 48: A higher-level introduction of the "probabilistic event attribution approach" is necessary for the readers to understand the concept. While technical details could refer to published papers, the general principle should be properly introduced.

We introduce further details of the "probabilistic event attribution approach" in lines 59-78 and 146-159.

Line 55-56: On what temporal horizon? This point is not clear at all. After reading the entire paper, it appears to me that the warming scenarios (1.5, 2.0, or 3.0 C) is defined regarding the reference year 2100. However, the simulated results in 2015 are discussed. This setting is problematic since the emission changes in each scenario are not linear across the century, what does it mean when comparing the first few years? The starting and ending point of the simulations are not clearly stated. The use of those simulations is not adequately justified.

We added SST anomalies of 1.5, 2.0, or 3.0 °C warmer climate (relative to the preindustrial) scenarios at the end of the 21st century taken from the RCP experiments of the CMIP5 AOGCMs for the observed 2006-2016 SST (HaISST) data. By using these experiments, we can investigate changes in droughts and fires when 2015-like El Nino events occur at the given future warmed levels, such as the 1.5°C and 2.0°C goals of the Paris Agreement. The computing cost of AGCM is lower than AOGCMs, which enables us to perform large ensembles (100 members for each experiment) that are necessary to estimate changes in the probabilities of extreme events. Please see lines 79-95 and 160-188 for more details.

Line 64: It will be helpful to state clearly how many ensemble members for each scenario, covering which period, based on what emission trajectory.

We state the ensemble numbers (100 for each experiment), period (11 or 10 years) and emission scenarios in section 3.

Line 77: "Although socio-economic factors are important for fire activities, we only examine the effects of climate change in this study." "climate change" here refers to simulated climate given different forcing scenarios, which by definition accounts for socio-economic factors. Maybe, the authors are referring to local land-use change impact? It is important to make those distinctions to make sense of the results.

We rewrote the sentence to "Although conversions of forest and peatlands to agriculture and plantations of oil palm are also important factors for fire activities (Marlier et al., 2013, 2015; Kim et al., 2015), we do not examine the effects of land use change in this study. " [lines 104-105]

Section 2. Compared to the papers cited here, I do not see any new contribution from this

section. It just shows what has been done, without any new data or insight.

Our main results are that we combine those empirical functions with the estimated probability functions of droughts based on the large ensembles to assess the historical and future warming effects on the probabilities of fire and emissions. We do not say that section 2 presents new results, but we do show the empirical functions here in order to use them in the following sections. [lines 114-129 and 215-221]

Line 118: "Please note that both the Hist and Nat ensembles have the same spatial SST patterns as the 2015 El NinÌC\* o event". How so?

Please see lines 146-159 for the details regarding the experimental design.

Line 124: If the simulation periods are 2006-2015, how come are they future simulations???

As mentioned above, we simulated extreme events when 2015-like El Nino events occur in the given future warming levels. [lines 18-22, 79-95 and 160-188]

Line 136: Why is showing sea ice (Fig. 5) relevant to this study? Similarly, it is not clear about the role of results showing in Figure 4 to Figure 7.

This is the second paper in which the 3°C runs of the HAPPI project have been performed. Although the first paper (Lo et al. 2019, using the different AGCM) briefly described the experimental design, it could not show the SST and ice patterns due to the limited space of paper. Therefore, we wanted to show the SST and sea ice patterns in Figs. 4-5 of the original manuscript. However, it is not necessary to include these figures in the main text. We have moved those to the supplementary material (Supplementary Figs. 2-4).

Figures 6 and 7 of the original manuscript (Figs. 4-5 of the current manuscript) are necessary in the main text to explain that the differences in the described SST between Niño 3.4 and the 30°S-30°N ocean clearly affect the vertical motion and precipitation anomalies in the EA region. Fig. 7 (Fig. 5 of the current version) is used to investigate why the precipitation anomalies of 1.5°C are slightly larger than those of 2.0°C. [lines 195-203]

Line 169: Are you comparing fire and fossil fuel emissions from Japan? Not clear.

We included this comparison because it was highlighted that the fire emissions of EA exceeded the fossil fuel emissions from Japan when the massive 2015 fire event occurred (e.g., Field et al. 2016). However, readers other than Japanese readers may be not interested in this comparison. Therefore, we have omitted this paragraph.

**Line 192-194: By now, it is still difficult to understand what do they mean. Line 192-212: The conclusion is far reached.**

We hope that the additional explanations of the probabilistic event attribution approach and the experimental design mentioned above help you to understand our results and why we have reached these conclusions.

**Line 197: "somewhat"?**

We rephrased "somewhat" to "tended to increase" [line 259].

**Reply to the reviewer #2**

Thank you very much for your helpful comments. Based on your comments, we have improved the manuscript.

**General comments:**

The authors present a policy-relevant study of changes in precipitation deficits and fire metrics in equatorial Asia (EA) during a 2015-like El Niño event and in a decadal average sense at 1.5\_, 2\_, and 3\_C warming levels. Results are based on a factual counterfactual probabilistic event attribution approach in a MIROC5 AGCM large ensemble framework. The following questions are explored:

• Did historical climate change increase the probability of the 2015 event?

• How will probabilities of drought, fire and fire emissions change when a major El Niño event similar to the one in 2015 occurs in 1.5\_C, 2.0\_C, 3.0\_C climate.

Authors find that historical anthropogenic forcing has increased the likelihood of a drier than-2015 El Niño-driven precipitation deficit in the EA from 2% to 9%. At 1.5\_C of warming, a drier event is 82% likely to occur. At 2\_C of warming, the probability of a drier event drops to 67% likely (for reasons that are not entirely clear) but increases to 93% at 3\_C of warming. This increased risk of drier conditions during El Niño events has ramifications for burned area extent, CO2 and particulate emissions in the regions.

The paper could be a valuable addition to the current body of literature on extreme events in warmer climates and the figures presented are clear and easy to interpret. There is care taken to connect findings to policy considerations whenever possible, particularly possible underestimates of EA CO2 emissions under climate change scenarios. However, the interesting findings would benefit from additional detail on the method, model experimental framework, relationships between relevant processes, and, most critically, on the value-add gained by using a 100-member large ensemble as opposed to 10-member ensembles. Addressing the specific comments below should more than adequately clarify and strengthen conclusions.

Thank you very much for the helpful comments. Please see the section below for our detailed responses to your specific comments.

**Specific comments:**

I recommend omitting the phrase: "the year" from the title.

Thank you. We changed the title to "Historical and future anthropogenic warming effects on droughts, fires and fire emissions of  $CO_2$  and  $PM_{2.5}$  in equatorial Asia when 2015-like El Niño events occur".

Abstract On what is the statement "caused" based on? [L14] What is the dry season in Equatorial Asia (i.e., in terms of months)? [L15] The acronym PM2.5 is not yet defined [L21]

Based on previous studies and the analyses of Figs. 1-3 (e.g., Fig. 3c shows the -0.89 correlation between the Nino3.4 SST and precipitation anomalies), we stated "caused", but this terms may be too strong. We rephrased it to "contributed to". [line 15] The dry season is June-November. [line 16]

We define acronym PM2.5 in lines 23-24.

Introduction It would be helpful to explicitly define when EA dry season occurs: : : Is the EA region the same as the SEA SREX region? If not, the specific latitude and longitude boundaries should be given. [L32-34]

We define the EA dry season (June-November) in line 37. We apologize that the EA region shown in the original Fig. 1 was incorrect. Actually, we use the definition of the EA region of GFED4s. We show the EA region in Fig. 1g and explain it in the caption of Fig. 1 and line 37.

Could you explain the ENSO phenomenon and how and why it "enhanced severe drought" to help guide readers? How did the 2015 drought compare to other ENSO events? This can be done through an explanation of the relationship between Walker circulation and convection and by including relevant citations on how the 2015 event compares to other El Nino in terms of effect on EA climate. [L32-34]

We explain the relationships between the 2015 El Niño and drought in lines 36-44 and 139-145.

Is the whole EA region considered tropical peatland? Can that qualification be defined (i.e. what is tropical peatland) and can it be explained why the region is susceptible to biomass burning? [L33]

We have improved these sentences as follows: "Parts of the EA region are tropical peatlands that contain tremendous amounts of soil organic carbon (Page et al., 2011) and huge biomass

(Baccini et al., 2012, 2017; Saatchi et al., 2011). Coupled with anthropogenic land-use change (e.g., expansion of oil palm plantations on peatlands), the severe drought increased fire activities in forests and peatlands" [lines 45-47]

**Can you elaborate more on the findings of Lestari et al. 2014? It will help readers understand how this new study extends the findings. [L43]**

I also recommend the following edit: "We use a probabilistic event attribution approach similar to Lestari et al. (2014), but our results are based on 100-member large ensembles of the MIROC5 AGCM with and without anthropogenic warming as opposed to 10-member ensembles." Then, a statement should be made about why the large ensembles were necessary and important to use. Justifying and highlighting the importance of large ensembles is a key point, especially for a special journal edition dedicated to large ensembles. [L48-50]

We explain the probabilistic event attribution approach, the difference between Lestari et al. (2014) and the present study, and why 100 member ensembles are necessary in lines 54-95 and 263-266. Because the 10 member ensembles of Lestari et al. (2014) are too small to examine how historical climate changes affected the probability of extreme events, we use 100 member ensembles [lines 74-76].

Question 1: Did historical climate change increase the probability of the 2015 event? How is historical climate change defined? Is it with respect to a certain base period? How is "change" defined in the presence of natural variability? [L47]

Can you elaborate on the probabilistic event attribution approach used in this study? It will help readers who are not familiar with detection and attribution techniques understand the opportunities and limitations of these approaches. How were they used in these cited papers? What were some of the key findings? [L48-54]

Historical climate change increased the probability of the 2015 drought event. We explain more details of the event attribution approach in lines 59-78 and 146-159, which should help readers understand how we define "historical change". On the other hand, we shorten the text mentioning the previous event attribution papers to prevent the paragraph from being too long [lines 76-78].

Question 2/3: Could these two sections be combined into one section about risk at 1.5, 2.0, and 3.0\_C warming? [L55]

We combine those paragraphs regarding the risks at 1.5, 2.0, and 3.0°C warming into one paragraph. [lines 79-95].

Just a small comment but can the connection between the initiation of the HAPPI project and the Paris agreement be smoothed out a little? Was the HAPPI project initiated in response to or to inform the Paris agreement? [L56-61]

The HAPPI project was initiated in response to the Paris agreement. [line 87]

In regards to "Although socio-economic factors (e.g., conversions of forest and peatlands to agriculture and plantations of oil palm) are also important for fire activities (Marlier et al., 2013, 2015; Kim et al., 2015), we only examine the effects of climate change in this study." Does this mean land-use change is not considered? What are the relevant "effects of climate change" on these events (i.e. warmer mean temperature, circulation changes, changes in ENSO?)

We rewrote those sentences to "Although conversions of forest and peatlands to agriculture and plantations of oil palm are also important factors for fire activities (Marlier et al., 2013, 2015; Kim et al., 2015), we do not examine effects of land use change in this study." [lines 104-105]

Empirical functions: Can you elaborate on your observational dataset choices? Why did you choose the reanalysis products you use? How are the enhanced fire fraction, fire CO2 emissions and fire PM2:5 emissions computed in the Global Fire Emissions Database?

ERA Interim reanalysis (ERA-I) data (Dee et al., 2011) are used for temperature and vertical circulation. GPCP is precipitation data that merge rain gauge stations, satellites, and sounding observations to estimate monthly rainfall on a 2.5-degree global grid from 1979 to present. Because these datasets have been used by an enormous number of atmospheric circulation studies, we also analyze these data. [lines 39-40]

By combining satellite information on fire activity and vegetation productivity, GFED4s provide monthly burned area, fire carbon and dry matter (DM) emissions. We can also compute aerosol emissions by multiplying DM by the provided factors. [lines 110-114]

Could a figure demonstrating the relationship between burned area, CO2 emissions, and particulate emissions be included? Is there a linear relationship between burned area and emissions?

Supplementary Figure 1 shows the relationships between fires and fire emissions in the EA area

of the GFED4s during 1997-2016. Clear linear relationships are shown. [line 114]

**What are the empirical functions used for in this study?**

We use the relationships in Figs. 2a-c as the empirical functions to estimate fire and emissions from the simulated precipitation. [lines 119-129 and 214-221]

Model simulations: Can you provide a further description of the MIROC5 AGCM? I.e. what is the horizontal resolution of the atmosphere? What observed SSTs specifically were used, particularly for the "natural" SST? How was the "long-term anthropogenic signal" defined and removed?

The MIROC5 AGCM has a 160 km horizontal resolution [line 132]. We used the HadISST data for the observed SST [lines 133-134]. We explain the long-term signal of SST and the Nat SST in lines 153-159.

Most importantly, what fire model is used? How is it related to the land surface state and coupled to the atmosphere? What triggers a fire in the model? How are CO2 and PM2:5 concentrations determined for a given event? [L103-121]

The MIROC5 model has no fire module. Therefore, we used the empirical functions of Fig. 2 to estimate fire and emissions from the simulated precipitation. [lines 128-129 and 214-221]

**What are the "corresponding standard deviation values"? [L108]**

Here, the observed  $\triangle P$  and  $\triangle \omega 500$  are divided by their own standard deviation values. The  $\triangle P$  and  $\triangle \omega 500$  of each ensemble member are also divided by their own standard deviation values. [lines 134-136]

Throughout the study, the descriptions of the figures are a little brief. In this case: "The precipitation and vertical motion anomalies are closely related to the Nino 3.4 SST (an index of El Niño Southern Oscillation) in the observations, and the MIROC5 model represents these relationships well (Figs. 3c-e)." How are they related (i.e., subsidence and reductions in rainfall during an El Niño)? What does "represent these relationships well" mean (i.e., significantly

**correlated with observations)? [L108-110]**

We have improved the descriptions in lines 139-145:

"The precipitation and vertical motion anomalies are closely related to the Nino 3.4 SST (an index of El Niño Southern Oscillation) in the observations (correlations are -0.89 and 0.76, respectively) (Figs. 3c-d). There is also a high correlation value between  $\Delta P$  and  $\Delta \omega 500$  (-0.87) (Fig. 3e). It is suggested that El Niño (La Niña) accompanies descending wind (ascending wind) in the EA area (Fig. 3d), leading to negative (positive)  $\Delta P$  (Figs. 3e and 3c). The MIIROC5 model well represents these relationships between Niño 3.4,  $\Delta P$  and  $\Delta \omega 500$  in the observations (Figs. 3c-e), i.e., the regression lines of MIROC5 in Figs. 3c-e are close to those in the observations."

How were "prescribed long-term warming anomalies in SST" defined? From Figure 4, I can see that there are spatial differences in warming, where do they come from? These details are likely important to the overall interpretation of the results and it would benefit the reader not to have to search for methodological descriptions in other studies or elsewhere in the paper. [L125-126]

We explain more details of the experimental designs of future simulations and how to add the SST anomalies in Fig. 4 to the observed data in lines 79-95 and 160-188.

The colorbar seems to be saturated in the bottom panel of Figure 4 over much of the Northern hemisphere, could the scale be adjusted to accommodate the 3\_C mean difference? Is there a difference between the respective Figure 5 top and middle panels?

We changed the color scale in Supplementary Fig. 2. Supplementary Fig. 4 shows the sea ice differences.

Could you detail how these results were reached using the cumulative density functions? Particularly, how did the use of large ensembles affect the results? Does the "chance of exceedance" change with fewer members? [L154-158]

To help readers understand our results, we add "2% (1-4%) in Nat to 9% (6-14%) in Hist", "(in the 1.5°C and 2.0°C runs)" and "(in the 3.0°C runs)" in lines 206, 211 and 213.

The large ensemble simulations enable us to estimate the probabilities of drought exceeding

the observed value [line 204]. We cannot robustly examine the probabilities using only the 10 members (10 samples) of Lestari et al. (2014) [lines 74-75].

Can you comment on why the chance of precipitation reduction exceedance more probable in the 1.5\_ scenario than the 2\_ scenario? [L157]

We explain this in lines 195-203.

"2015 CO2 emission of Japan due to fossil fuel consumptions" is a missing a citation [L168-169]

We included this comparison because it was highlighted that the fire emissions of EA exceeded the fossil fuel emissions from Japan when the massive 2015 fire event occurred (e.g., Field et al. 2016). However, readers other than Japanese readers may be not interested in this comparison. Therefore, we have omitted this paragraph.

I am sorry I may have missed something, but what is the AIM/CGE model used for? Was it introduced in the methods section? [L179-180]

AIM/CGE is the one of integrated assessment models (economic models) that produced emissions data for the SSP scenarios of CMIP6. In this paragraph, we suggest that additional fire CO2 emissions due to climate change should be considered in emission scenarios that are used for the next CMIP7 future projection experiments. We have improved this paragraph [lines 228-248].

**Reply to the reviewer #3**

Thank you very much for your helpful comments. Based on your comments, we have improved the manuscript.

General comments: This study examines global warming impacts on fire activities, focusing on the burned area and fire emissions of CO2 and PM2.5 over equatorial Asia. Considering June-November 2015 when a strong El-Nino induced a large decrease in precipitation over the area, the authors examine changes in the probabilities of droughts and fire activities due to anthropogenic influences using the MIROC5 AGCM large ensemble (100 members) simulations. They find increased probabilities of the droughts and fire activities as global warming become stronger. In particular, they show that 3.0 degree warming that represents the current mitigation policies would bring severe droughts and increased fire activities due to the intensified El Nino at near 100% chance. I find this paper overall well written, providing interesting and policyrelevant results. However, there are a few issues, mostly related to uncertainty factors, which need to be improved through revision.

Thank you. Please see our replies to the following comments.

Major points: 1. Model dependency: It would be useful to discuss limitations of the atmospheric model experiments and its possible impacts on the results. Particularly, precipitation changes in the future warming simulations are shown to be critical for determining changes in fire burned area and CO2 and PM2.5 emissions (Fig. 8), but atmospheric models tend to have large biases in precipitation over the Tropics partly related to the omission of air-sea coupling. It seems that normalized precipitation is used to overcome this problem but some justification would be needed with showing precipitation bias of the model. In addition, future projections of precipitation look highly dependent on the SST change patterns (Fig. 4). Uncertainty in these SST change patterns needs to be discussed as well.

Unfortunately, MIROC5 is only one model that produced all the Nat, Hist,  $1.5^{\circ}$ C,  $2^{\circ}$ C and  $3^{\circ}$ C ensembles. Therefore, we cannot use the other HAPPI models for this study. With a simple bias correction (i.e., dividing precipitation anomalies by their standard deviation values), the MIROC5 model has very good hindcast skill regarding interannual variability in the EA-averaged  $\Delta$ P and  $\Delta\omega$ 500 (correlation values between the model and observations are ~0.9) [lines 134-145]. We also suggest that our future projections are consistent with previous studies that have analyzed the CMIP5 ensemble: Lestari et al. (2014) and Yin et al. (2016) also showed

that the coupled model ensembles of CMIP5 projected future drying trends and enhanced fire carbon emissions [lines 263-266]. We also add a caveat in lines 280-282.

2. New findings: New results compared to previous studies are not clearly explained, in particular, in view of Lestari et al. (2014). What advances have been achieved by increasing ensemble size? Adding more information on this would be helpful, such as how to construct ensembles and how uncertainty is assessed with the large ensemble simulations.

Although Lestari et al. (2014) showed anthropogenic effects on the historical trends of droughts, it is not clear how historical climate changes affected the *particular drought event of 2015*. Based on probabilistic event attribution, we investigated whether historical climate changes affected the 2015 event. Because the 10 member ensembles of Lestari et al. (2014) are too small to estimate the probabilities of extreme events, we use 100 member ensembles of Shiogama et al. (2014). The computing cost of AGCM is lower than AOGCM, which enables us to perform such large ensembles that are necessary for PEA. [lines 59-78 and 204]

Although Lestari et al. (2014) and Yin et al. (2016) showed increases in droughts and fires in the future projection ensembles of AOGCMs, it is not clear how future anthropogenic warming affects droughts and fire when 2015-like El Niño events occur in a future warmer climate. It is important to investigate changes in extreme events at 1.5°C and 2.0°C warming levels to inform stakeholders, as the Paris Agreement set the 2°C long-term climate stabilization goal and is pursuing 1.5 °C to reach stabilization (United Nations Framework Convention on Climate Change 2015), but Lestari et al. (2014) and Yin et al. (2016) did not perform such analyses. In this study, we examine how the probabilities of drought, fire and fire emissions of CO2 and PM2.5 would change when major events like the 2015 El Niño occur under 1.5°C, 2.0°C and 3.0°C warmed climates. [lines 79-95 and 263-266].

We added details of the experimental designs in section 3. We also explain how uncertainty is assessed with the large ensemble simulations below.

Also, the empirical relation between precipitation and fire activities is used to estimate future changes in fire activities and the authors consider its uncertainty somehow in their analysis. I think this part is important and more details needs be provided on its uncertainty ranges and associated impacts on main results. See my specific points below.

L141-144: This way of sampling looks important to capture uncertainty arising from internal variability, and showing resulting spreads in P and omega responses in Fig. 7 would be

interesting. Also, it would be useful to explain here how to construct CDF using 1000 samples and estimate probabilities exceeding the observed value and its 10-90% confidence intervals. L161: "1000 random samples of the regression factors in Eq. 1". Please provide details given its importance. Also see my major comment.

We explain how to construct the CDFs and estimate the uncertainty ranges by using the large ensembles and the resampling techniques as follows.

"We also estimate the 10%-90% confidence intervals of the fitting curves by applying a 1000time random sampling of the observed data: we randomly resample 20-year samples from the original 20-year (1997-2016) data and compute a and b; we repeat the random resampling process 1000-times; we consider that the 10%-tile and 90%-tile values of the 1000 regression lines indicate the 10%-90% confidence intervals." [123-126]

"Here, we use the cumulative histograms of  $100 \times 10=1000$  samples of  $\triangle P$  to estimate the probabilities of  $\triangle P$ . The values in parentheses indicate the 10-90% confidence interval estimated by applying the 1000-time resampling: we randomly resample  $100 \times 10$  data from the original  $100 \times 10$  samples of  $\triangle P$  and compute the probabilities of drought exceeding the 2015 observed value; we repeat the random resampling process 1000-times and consider the 10%-tile and 90%-tile values of the 1000 estimates of probability as the 10-90% bounds." [lines 206-210]

"We consider uncertainties by combining randomly resampled  $\triangle P$  and resampled regression factors of Eq. 1: (i) we compute the regression factors of Eq. 1 using randomly resampled data (the same as the process used to estimate the uncertainty ranges of the regression lines); (ii) we randomly resample 100×10 data from the original 100×10 samples of  $\triangle P$ ; (iii) we use the regression factors of (i) and the 100×10  $\triangle P$  samples of (ii) to compute the 1000 estimates of fire or emissions and estimate the probability of exceeding the observed values; (iv) the processes of (i)-(iii) are repeated 1000-times; and (v) the 10%-tile and 90%-tile values of the 1000 estimates of the probabilities of exceeding the observed values are considered to be the 10-90% bounds." [lines 215-221]

3. Implications: The last part on implications is rather confusing and hard to follow. I would suggest rephrasing it for better understanding. For example, it is unclear what are exactly

compared between MIROC5-based estimations and diverse SSP scenarios: fire CO2 emissions due to climate change versus land use CO2 emissions? From this comparison, the authors seem to suggest that additional fire CO2 emissions due to climate change should be considered in SSP scenarios, but this interpretation is not that clear at the present form. I am wondering if it can be made more specific by suggesting how much increase in CO2 emissions should be added, for example.

We improved this paragraph [lines 228-248]. Currently, it is not easy to compute fire CO2 emissions due to future drying in AIM/CGE because we have to develop a new fire module considering climate change effects on fire for AIM/CGE. The development of such a new module is an issue for subsequent CMIP7 activity.

Specific points: Title: "year 2015" sounds a bit strange to be connected with "future" warming effects. How about saying "2015-like" or similar instead.

We changed the title to "Historical and future anthropogenic warming effects on droughts, fires and fire emissions of CO2 and PM2.5 in equatorial Asia when 2015-like El Niño events occur"

L87-88: How is the EA box selected? I think it can be adjusted (e.g., narrower in zonal direction) to better capture the P decrease area. Or it doesn't matter since only land is considered? Please clarify this.

We apologize that the EA region shown in the original Fig. 1 was incorrect. Actually, we use the definition of the EA region of GFED4s. We show the EA region in Figure 1g and explain it in the caption of Figure 1 and lines 36-37.

L91-92: In line with "precipitation anomalies and accumulated water deficits", wouldn't it be better to use accumulated precipitation like SPI?

We use the June-November mean precipitation, which is the accumulated precipitation during the dry season divided by the period length. Therefore, we substantially use the accumulated precipitation anomalies.

L93, L108: "divided by standard deviation". Can we assume normality for precipitation and omega anomalies? Area averaged 6-month mean values might be okay but a quick check would be useful.

The following figure shows the cumulative distribution functions of normalized precipitation and omega anomalies and those of Gaussian distribution. It is suggested that the area-averaged 6-month mean values have a Gaussian distribution.

L115: I would suggest providing more details on how "long-term anthropogenic signals were removed" as SST patterns are important for determining precipitation responses to El Nino.

Anthropogenic SST changes were estimated by taking the ensemble mean differences between the all-forcing historical runs and the natural-forcing historical runs of the CMIP5 ensembles. The multi-model averaged anthropogenic signals were subtracted from the HadISST data, and the Nat sea ice was estimated by using empirical functions between observed sea ice concentrations and surface temperature. [lines 153-159]

L122: "100 member ensembles during 2006-2015 with 1.5 degree and 2.0 degree warming". Do it mean that ensemble runs are performed only for Plus15 and Plus20 or there are 100-member HIST runs for 2006-2015 as well?

We performed 100 member runs of 2006-2016 for each of Hist, Nat, Plus15 and Plus20 [lines 158-159, 160-161 and 166-167]. The Plus30 are 100 member runs of 2006-2015 [line 176].

L150-153: Why stronger El Nino (and P responses) are simulated in 1.5 degree warming simulations than 2.0 degree ones? Some discussion needs to be provided. Does it occur in all

1000 samples? Do other HAPPI models share this or is this a characteristic of MIROC5?

All the HAPPI models share the SST anomalies that were taken from the CMIP5 model ensembles [lines 160-175]. It is not clear why the ensemble average of the CMIP5 RCP2.6 runs (i.e., the prescribed SST anomalies of the 1.5 °C runs) has a larger SST difference between the Niño 3.4 region and the tropical ocean mean than that of the weighted sum of RCP2.6 and RCP4.5 (the 2.0 °C runs) [lines 201-203]. The differences in the prescribed SST warming contrasts between the 1.5°C and 2°C runs cause the difference between the blue and green CDFs in Fig. 6a [195-212].

**L163-164: Please explain how to assess significance of this change.**

By comparing the uncertainty bounds of future changes with the uncertainty bounds of Hist and Nat, we assess the significance of changes.

**L169: Why is the emission of Japan used as reference here?**

We included this comparison because it was highlighted that the fire emissions of EA exceeded the fossil fuel emissions from Japan when the 2015 massive fire event occurred (e.g., Field et al. 2016). However, readers other than Japanese readers may be not interested in this comparison. Therefore, we omitted this paragraph.

L172: First sentence. This needs to be mentioned clearly above and also in figure captions to avoid confusing.

L172-188: This paragraph and Fig. 9 are hard to follow with many skips and limited explanations. Please consider rephrasing it. See my major comment above.

We have improved our explanations in lines 228-248.

L196: "82%, 68%, and 93%". Please add uncertainty ranges or indicate these are ensemble means or medians. Same for L204.

We add the uncertainty ranges in lines 254, 256 and 261-263.

L199-202: Model dependency issue is here. How representative is MIROC5 projected

precipitation in the future? Any comparison with other models would be useful. See my major comment above.

Please see our responses to your major comments.

**L205: "additional changes". Are these significant?**

Although the differences between 2.0°C and 3.0°C are not statistically significant for the burned area and the CO2 and PM2.5 emissions, the 50th percentile values of probabilities exceeding the 2015 observations first reach approximately 100% in the 3.0°C runs. [lines 270-272]

L209: "modifying fire CO2 emissions scenarios". Can authors suggest how much modification is needed? See my major comment above.

Currently, it is not easy to compute fire CO2 emissions due to future drying in AIM/CGE because we have to develop a new fire module considering climate change effects on fire for AIM/CGE. The development of such a new module is an issue for subsequent CMIP7 activity.

Fig. 1: Line 342: "left panels" should be "right panels".

We corrected this issue in the caption of Fig. 1.

Fig. 2: There seems to be a stronger case than 2015, perhaps 1998? Where is 1982 that has also a stronger P decrease in Fig. 3? It may affect fitted curves.

We apologize that the explanations of the original manuscript were not corrected. Although we used 1979-2016 GPCP data, GFED4s covered only 1997-2016. Thus, Fig. 2 shows the scatter plots between precipitation and GFED4s during 1997-2016, not 1979-2016. We corrected this mistake in lines 116-119 and the caption of Fig. 2. Therefore, 1982 is not included in this figure. The 1997 case (the 1997-1998 El Nino) is stronger than 2015 case. We indicate 1997 in Fig. 2.

Fig. 3: Indicating 2015 case in time series and scatter plots would be useful. Is there any underestimation or overestimation by models in P and omega responses?

We indicate the 2015 case in Fig. 3. The model estimates P and omega responses well [lines 136-145].

Fig. 6: It's not clear why difference from NAT is shown even for future changes. Is this for 2015 or using all years?

We show differences from NAT because the mixing of differences from Hist for future changes and that from NAT for Hist may confuse readers. These figures are for 2015 (caption of Fig. 4).

Fig. 7: Is this also for 2015? Related to my major comment on model dependency issue, are these are supported by other coupled models?

This figure is for 2015. Please see our responses to your major comments.

Fig. 9: Difficult to understand. How is the CDF of CO2 emissions (red curves) estimated? Are these CO2 emissions only due to increased fire over equatorial Asia?

We improved the descriptions in lines 228-248.

**Historical and future anthropogenic warming effects on droughts, fires and fire emissions of CO2 and PM2.5 in equatorial Asia when 2015-like El Niño events occur**

5 Hideo Shiogama1, 2, Ryuichi Hirata1, Tomoko Hasegawa3, Shinichiro Fujimori4, Noriko Ishizaki1, Satoru Chatani1, Masahiro Watanabe2, Daniel Mitchell5, Y. T. Eunice Lo5

1National Institute for Environmental Studies, 16-2 Onogawa, Tsukuba, Ibaraki 305-8506, Japan 2Atmosphere and Ocean Research Institute, University of Tokyo, 5-1-5 Kashiwanoha, Kashiwa, Chiba 277-8564, Japan

3College of Science and Engineering, Ritsumeikan University, 1-1-1 Noji-higashi, Kusatsu, Shiga 525-8577, Japan 4Department Environmental Engineering, Graduate School of Engineering, Kyoto University, Kyoto 615-8540 Japan 5School of Geographical Sciences, University of Bristol, University Road, Bristol BS8 1SS, United Kingdom

Corresponding author: Hideo Shiogama (shiogama.hideo@nies.go.jp)

- 15 Abstract. In 2015, El Niño contributed to severe droughts in equatorial Asia (EA). The severe droughts enhanced fire activities in the dry season (June-November), leading to massive fire emissions of  $CO_2$  and aerosols. Based on large event attribution ensembles of the MIROC5 atmospheric global climate model, we suggest that historical anthropogenic warming increased the chances of meteorological droughts exceeding the 2015 observations in the EA area. We also investigate changes in drought in future climate simulations, in which prescribed sea surface temperature data have the same spatial patterns as the 2015 El
- 20 Niño with long-term warming trends. Large probability increases in stronger droughts than the 2015 event are projected when events like the 2015 El Niño occur in the 1.5°C and 2.0°C warmed climate ensembles according to the Paris Agreement goals. Further drying is projected in the 3.0°C ensemble according to the current mitigation policies of nations.

We combine these experiments and empirical functions among precipitation, burned area, and fire emissions of  $CO_2$  and fine (<2.5 micrometers) particulate matter ( $PM_{2.5}$ ). Increases in the chances of burned areas and the emissions of  $CO_2$  and  $PM_{2.5}$

25 exceeding the 2015 observations due to past anthropogenic climate change are not significant. In contrast, there are significant increases in the burned area and CO2 and PM2.5 emissions even if the 1.5°C and 2.0°C goals are achieved. If global warming reaches 3.0°C, as is expected from the current mitigation policies of nations, the chances of burned area, CO2 and PM2.5 emissions exceeding the 2015 observed values become approximately 100%, at least in the single model ensembles.

We also compare changes in fire  $CO_2$  emissions due to climate changes and the land-use  $CO_2$  emission scenarios of five 30 shared socioeconomic pathways, where the effects of climate change on fire are not considered. There are two main implications. First, in a national policy context, future EA climate policy will need to consider these climate change effects regarding both mitigation and adaptation aspects. Second, the consideration of fire increases would change global  $CO_2$ emissions and the mitigation strategy, which suggests that future climate change mitigation studies should consider these factors.

**35 1 Introduction**

In 2015/2016, a major El Niño event (strongest since 1997/1998) enhanced severe drought in equatorial Asia (EA, the area denoted in Fig. 1g) during the dry season (June-November). Figures 1a-c indicate the observed June-November 2015 mean anomalies in surface air temperature ( $\Delta$ T), vertical pressure velocity at the 500-hPa level ( $\Delta \omega_{500}$ ) and precipitation ( $\Delta$ P) relative to the 1979-2016 averages. ERA Interim reanalysis (ERA-I) data (Dee et al., 2011) are used for  $\Delta$ T and  $\Delta \omega_{500}$ , Global

[revised manuscript text omitted]

---

## Author Response (AR2)

**Reply to the reviewer #3**

Thank you very much for your helpful comments. Based on your comments, we have improved the manuscript.

- Re: We explain the relationships between the 2015 El Niño and drought in lines 36-44 and 139-145. We apologize that our comment wasn't clear; the description of the 2015 El Niño event in the introduction should not be done in terms of Figure 1, it should be done in terms of relevant literature.

Recommendations:

Stockwell, C. E., Jayarathne, T., Cochrane, M. A., Ryan, K. C., Putra, E. I., Saharjo, B. H., ... & Stone, E. A. (2016). Field measurements of trace gases and aerosols emitted by peat fires in central Kalimantan, Indonesia, during the 2015 El Niño. Atmospheric Chemistry & Physics, 16(18).

Liu, J., Bowman, K. W., Schimel, D. S., Parazoo, N. C., Jiang, Z., Lee, M., ... & O'Dell, C. W. (2017). Contrasting carbon cycle responses of the tropical continents to the 2015–2016 El Niño. Science, 358(6360), eaam5690.

Santoso, A., Mcphaden, M. J., & Cai, W. (2017). The defining characteristics of ENSO extremes and the strong 2015/2016 El Niño. Reviews of Geophysics, 55(4), 1079-1129.

Thank you for your advice. By citing these papers and others, we explain the relationships between the 2015 El Niño, Walker circulation, drought and emissions in lines 36-41 and 47-48.

- It's challenging to follow the added discussion of Lestari et al. 2014 because it comes before the discussion of what is being done [L84]. Additionally, [L72-74] and [L84] are somewhat redundant, could they be combined and could the description of what was and wasn't done in Lestari et al. 2014 be moved to the discussion section?

These two paragraphs including [L72-74] and [L84] are different form each other, because they explain the studies of the attribution of historical changes and the future projections, respectively. Before these two paragraphs, we describe that Lestari et al. (2014) investigated both the historical trends and future projections in lines 52-56.

[L84] just denoted that Lestari et al. (2014) and Yin et al. (2016) did not analyze droughts at specific warming levels. We omit this sentence and add "transient" in line 75 and "also" in line 77.

We suppose that [L72-74] was [L74-75] "Because the 10 member ensembles of Lestari et al. (2014) are too small to estimate probabilities of extreme events, we use 100 member ensembles of Shiogama et al. (2014)". We omit this sentence. Instead, we write "We use the 100-member PEA ensembles of

MIROC5 (Shiogama et al. 2014)" in line 73 and "The 10 member ensembles of Lestari et al. (2014) were too small to estimate probabilities of droughts. Our large ensemble simulations enable us to estimate the probabilities of drought exceeding the observed value" in lines 206-207.

Empirical Functions
- The descriptions of Figure 1 and the datasets used should be in this section.

We move those to lines 105-111.

- L135: "the observed △P (△ω500) is divided by their standard deviation value." -> …△P and △ω500 are divided…

We change those in line 137.

Results
- It would still be interesting to determine why the 2˚C pathway leads to a less intense SST anomaly, more precipitation, and therefore less change in the fire statistics. Does it have anything to do with the weighted sum? My understanding is that the weighted sum is applied to the RCPs and not the prescribed SSTs, is that correct? [L169]

The weighted sum is applied for both the concentrations and the prescribed SSTs [lines 170-174]. Therefore it is necessary to reveal the reasons of differences in "El-Nino like warming patterns" between the RCP scenarios to understand the why the 2˚C pathway leads to a less intense SST anomaly. Although it is an interesting and challenging research topic, it beyond the focus of this paper.

- I notice the uncertainty range in changes in burn area fraction, CO2 and PM2.5 reach 0% change, but do not go below. Does this indicate there is no chance for smaller-than-observed fire in any scenario? [L223-225]

For example, 0%-5% chances of exceeding the observed value indicate 95%-100% chances for smaller-than-observed fire.

---

## Author Response (AR3)

**Dear Prof. Nicola Maher,**

Thank you for your useful comments. We checked and improved the manuscript.

Line 20 - 'of' stronger droughts
We changed 'in' to 'of' in line 20.

Line 23 - 'among' is not the correct word here, perhaps 'of' is better
Line 23-26 is difficult to understand can you reword?
We reworded those lines as the following in lines 23-26.
"We use observation-based empirical functions to estimate burned area, fire $CO_2$ emissions and fine (<2.5 μm) particulate matter ($PM_{2.5}$) emissions from these simulations of precipitation. There are no significant increases in the chances of burned areas and the $CO_2$ and $PM_{2.5}$ emissions exceeding the 2015 observations due to past anthropogenic climate change. In contrast, even if the 1.5°C and 2.0°C goals are achieved, there are significant increases in the burned area and CO2 and PM2.5 emissions."

Line 29 – I think that this should this be climate change not climate changes? If you agree please check the entirety of the manuscript for this

We changed "climate changes" to "climate changes" in lines 30, 58 and 70.

Line 37 - 'a' weakening

"a" was inserted in line 37.

Line 38 - 'relate' should be 'relates' and perhaps 'drives' or 'corresponds' would be a better choice of word

We used "corresponds" (line 38)

Line 45 - 'loss' for economy

We rewrote the sentence to "large economic loss  (at least 16.1 billion USD for Indonesia) and significant impacts on ecology and human health (Taufik et al., 2017; World Bank 2016, Hartmann et al., 2018)." in lines 44-46.

Line 47 - 'the' largest

"the" was inserted (line 47).

Line 62 – two sets of large ensembles

Done (line 62).

Line 63 - 'are' historical

'is' was changed to 'are'.

Line 129 - 'consistent'

We corrected it (line 129).

Line 140 - 'apply' is not needed

We rewrote the sentence to "we apply the above normalization process" (line 140).

Line 144 - 'Should this be 'it is suggested' or should it be 'we show that'

'we show that' is used. (line 144)

Line 149 - , ' one with factual … and one with counter factual'

Done (line 149).

Line 157 - 'signal'

We used 'signal' (line 157).

Line 158 'using an empirical function that estimates x from y'

We rewrote the sentence to "using an empirical function that computes observed sea ice concentrations from surface temperature" line 158.

Line 200 - 'an' El Nino

We inserted 'an' (line 201).

Line 201 – define tropical ocean mean

It is defined in line 199.

Line 236 – define major El Nino year

We wrote "the year 2015 with the major El Niño" in line 237.

Line 244 - be specific do you mean 'fire emissions'

Line 244 should be line 246. We wrote 'fire emissions of $CO_2$' in line 248.

Line 266 – Add – These results agree well with Lestart and Yin who also …

Done (line 268).

Line 283 – This would read better as – 'Future work to compare … would be useful.'

We rephrased it to "A future work to compare multi model simulations using multiple estimates of warming patterns in SST would be useful." (line 286-287)

Caption F4 – What do you mean by omitted? Should this be subtracted?

"subtracted' is used.

In general when you say fire or estimates fine (e.g. line 130, and the figure captions) do you mean a specific aspect of fire? If so this should be specified in the text

We improved the following sentences:
Line 130 "burned area and fire emissions"
Line 197 "burned area and fire emissions of $CO_2$ and $PM_{2.5}$"

Line 271 "burned area"

Line 282 "burned area and fire emissions"

Caption of F2 "burned area"

Caption of F7 "fire emissions of $CO_2$"

Caption of Supplementary Figure 1 "burned area"

I additionally ask that you add the following information to the manuscript before publication. A line or two of discussion about why the probability is smaller for 2degrees of warming than 1.5degrees of warming (see line 215).

The reason of larger responses in 1.5 degrees than 2 degrees are already mentioned in lines 201-206.

A line of discussion on a similar topic for probabilities stated in lines 243-244, the probabilities are not linearly related to the SSP number. Can you suggest why?

We added "Please note that the year 2100 land-use $CO_2$ emissions are not linearly related to the SSP numbers, because the SSP numbers did not indicate radiative forcing levels." in line 244-245.

Best regards,
Hideo Shiogama